

# Equilibration of multitime quantum processes in finite time intervals

**Neil Dowling**[1⋆], **Pedro Figueroa-Romero**[2], **Felix A. Pollock**[1],
**Philipp Strasberg**[3] **and Kavan Modi**[1†]

**1** School of Physics & Astronomy, Monash University, Victoria 3800, Australia
**2** Hon Hai Quantum Computing Research Center, Taipei, Taiwan
**3** Física Teòrica: Informació i Fenòmens Quàntics, Departament de Física,
Universitat Autònoma de Barcelona, 08193 Bellaterra (Barcelona), Spain

⋆ neil.dowling@monash.edu , † kavan.modi@monash.edu

## Abstract

A generic non-integrable (unitary) out-of-equilibrium quantum process, when interrogated across many times, is shown to yield the same statistics as an (non-unitary) equilibrated process. In particular, using the tools of quantum stochastic processes, we prove that under loose assumptions, quantum processes equilibrate within finite time intervals. Sufficient conditions for this to occur are that multitime observables are coarse grained in both space and time, and that the initial state overlaps with many different energy eigenstates. These results help bridge the gap between (unitary) quantum and (non-unitary) statistical physics, i.e., when all multitime properties and correlations are well approximated by stationary quantities, which includes non-Markovianity and temporal entanglement. We discuss implications of this result for the emergence of classical stochastic processes from multitime measurements of an underlying genuinely quantum system.

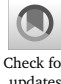

# 1  Introduction

Quantum processes allow the description of multitime statistics in quantum systems, providing information on the dynamics beyond the single time (quantum state) limit. Any quantum process can be represented as a single positive tensor $\Upsilon_{\mathbf{k}}$ [1, 2], that can be used to compute the expectation value of a multitime measurement,

$$\mathrm{tr}[\mathcal{A}_k \mathcal{U}_k \cdots \mathcal{A}_1 \mathcal{U}_1(\rho)] = \mathrm{tr}[\Upsilon_{\mathbf{k}} \mathbf{A}_{\mathbf{k}}^{\mathrm{T}}] =: \langle \mathbf{A}_{\mathbf{k}} \rangle_{\Upsilon} \ . \tag{1}$$

Here, $\mathbf{A}_{\mathbf{k}}$ too is a tensor encoding the sequence of $k$ singletime measurement operators $\{\mathcal{A}_k, \mathcal{A}_{k-1}, \ldots, \mathcal{A}_1\}$ (in the Schödinger picture) and $\Upsilon_{\mathbf{k}}$ is called the process tensor [1–3], which encapsulates the unitary dynamics ($\mathcal{U}_i$) and initial state ($\rho$). Both of the tensors have free indices at the times $\mathbf{k} := \{t_k, t_{k-1}, \ldots, t_1\}$, and are quantum combs [4]. These are depicted graphically in Fig. 1 (a), and will be more formally constructed later in this work. Such a description of quantum processes allows the characterization of temporal features such as the degree of non-Markovianity of a process [5, 6], the genuine multipartite entanglement in time [7], or when the statistics look classical [8, 9].

One can ask then, when such quantum processes look equilibriated? This is a foundational question of quantum statistical mechanics, concerned with how a thermal, or more generally a steady state, can arise from the picture of isolated quantum mechanics. There are a number of approaches to this research program, such as the celebrated Eigenstate Thermalization Hypothesis (ETH) which assumes that matrix elements of single time expectation values agree with their thermal value [10–18]. Equilibration, on the other hand, relies on rather minimal assumptions to show when expectation values look stationary on average. More specifically, for a quantum state in a large enough Hilbert space with many significant populations in the energy eigenbasis, realistic (coarse) observables look stationary on average [19–22].

Previous approaches to the task of deriving statistical mechanics from pure quantum mechanics predominately lie firmly within the single-time picture of quantum mechanics [13, 14, 23, 24]. However, this does not offer a complete picture, as generally there is a wealth of

hidden informational content in a quantum process $\Upsilon_{\mathbf{k}}$ compared to a state $\rho$, in the form of encoded temporal correlations. Our previous work Ref. [25] takes the first steps in addressing this problem, where it is shown which conditions lead analytically to a quantum process $\Upsilon$ equilibrating on average (over infinite times) to an equilibrium process $\Omega$, where all unitary dynamics is replaced with dephasing (as summarized in Fig. 1). This means that equilibration is stable to perturbation, and that multitime correlations look stationary on average over long times. Much like the single time results, this occurs for an effectively large initial state and a spatially coarse measurement, but interestingly it additionally requires that the multitime observable is coarse in *time* (i.e., that the number of measurements $k$ is not too large). This result, however, does not give any information on the time scales necessary for process equilibration, or the emergence of multitime features such as stochastic classicality.

Our results here and in Ref. [25] approach the question of equilibration from a new perspective. The advantage of considering the oft-neglected multitime setting, is that it exposes a wide plethora of phenomena of interest to the overarching question of the foundations of statistical mechanics. This includes why Markovianity is so prevalent in nature, and more generally why any multitime correlations should be well-approximated by stationary or thermal ones. For example, on the (microscopic) quantum level, a molecule will 'remember' if one applies laser pulse to it. However, chemically such a process is highly Markovian in practice. Yet, one theory clearly underlies the other. So how can one get from the microscopic theory to the (relatively) macroscopic? Measurement and interaction generally perturb quantum systems, so it is non-trivial to ask, what is the mechanism of the emergence of these multitime statistical properties, and how quickly do they appear? While quantum processes account for invasiveness of measurements, classicality emerges which is distinctly non-invasive [8,9,26]. Our work therefore helps address foundational questions *beyond thermalization and single-time equilibration*.

In this work we extend the results of Ref. [25] to show the equilibration of processes in finite time intervals and with degenerate energy levels. This means that, considering only the times between measurements within the interval $\Delta t_i \in [0, T_i]$, for large enough $T_i$ the processes $\Upsilon_{\mathbf{k}}$ and $\Omega_{\mathbf{k}}$ are indistinguishable. That is, under this constraint for coarse multitime observables $\mathbf{A}_{\mathbf{k}}$,

$$\langle \mathbf{A}_{\mathbf{k}} \rangle_{\Upsilon} \approx \langle \mathbf{A}_{\mathbf{k}} \rangle_{\Omega} \ . \tag{2}$$

These time intervals $T_i$ will be shown to be typically much smaller than recurrence times. Indeed, it is important that this is the case for the equilibration time scales to be meaningful. The infinite time intervals result of Ref. [25] readily implies the approximate equilibration for finite times equal to recurrence times, up to arbitrary accuracy. This is for any quantum processes evolving according to a finite dimensional, time-independent Hamiltonian, with arbitrary energy degeneracies and energy gap degeneracies. This is a generalization of the infinite time results, which relied on a technical assumption about energy gap degeneracies. Additionally, we also here give a general theorem of the equilibration of multitime geometric measures of quantum processes, showing the power of this result in comparison to single time equilibration. This may prompt further bounds on the equilibration of multitime properties, such as the time scales for a generic process to look Markovian or classical.

In section 2 we will introduce the process tensor formalism that describes quantum processes, together with the notion of instruments which correspond to arbitrary multitime measurements. Additionally we will recap the infinite times result of Ref. [25]. In section 3 we give our main result on the equilibration of processes in finite time intervals, together with an analysis of what time scales this will occur. Finally, in section 4 we show that process equilibration implies the general equilibration of any geometric measure of the multitime properties of a process, such as non-Markovianity, classicality, etc.

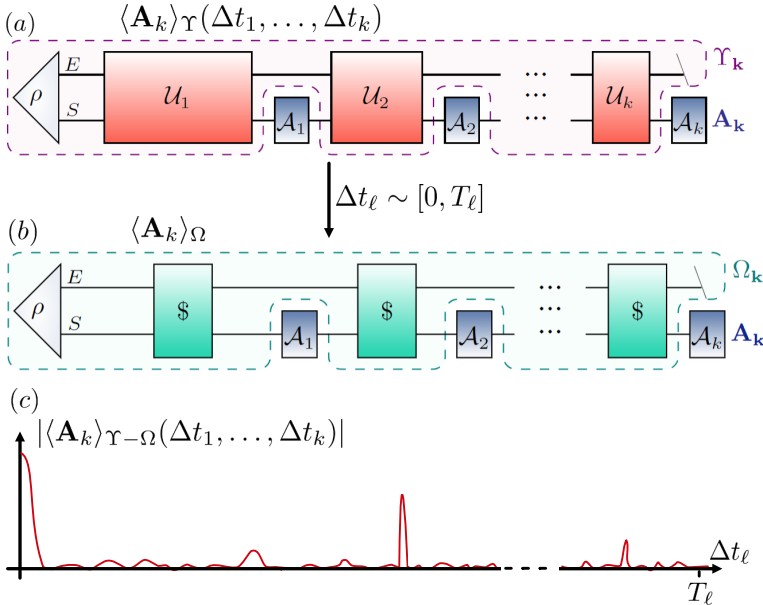

Figure 1: (a) An expectation value of the multitime instrument $\mathbf{A_k}$ over an arbitrary quantum process $\Upsilon$, which is dependent on times $\Delta t_\ell$ of unitary evolution $\mathcal{U}_\ell$. (b) If these times are sampled from a large enough ranges of time, $[0, T_\ell]$, then this expectation value is likely to be approximately time-independent, equal to the expectation value over the equilibrated process $\Omega$. (c) A visualization of the effect of process equilibration. Deviations from the stationary value of multitime correlations are rare and small, across the time ranges $\Delta t_\ell \sim [0, T_\ell]$ for $1 \leq \ell \leq k$. The conditions on $k$ and $T_\ell$ for this to occur can be seen in Eq. (29).

## 2 Preliminaries

Here we introduce the process tensor formalism from which our results are constructed from, and restate a main theorem from Ref. [25] from which this work generalizes to finite times and degenerate energy gaps.

### 2.1 Single-time instruments

Physical quantum transformations, including measurements, are in full generality described by a linear and completely positive (CP) map $\mathcal{A}$ that takes an input quantum state $\sigma$, to an output quantum state $\sigma'$. Both the input and the output states are density operators, but the latter is not necessarily normalized except when the map is deterministic, i. e., when $\mathcal{A}$ is further specified to be trace-preserving (TP).

Such a map admits a number of explicit representations, each useful is different circumstances [27, 28]. The operator sum or Kraus representation is given by

$$\mathcal{A}(\sigma) := \sum_{\alpha=1}^{n} K_\alpha \sigma K_\alpha^\dagger, \tag{3}$$

where if the representation is 'minimal', $n$ is equal to the rank of the map. A second representation that will be essential below, is the *Choi* state. For a map $\mathcal{A} : \mathcal{H}_b \to \mathcal{H}_c$ acting on an input $\sigma$, this is a matrix $\mathsf{A}$ such that

$$\mathcal{A}(\sigma) = \text{tr}_b[(\mathbb{1}_c \otimes \sigma^\mathsf{T})\mathsf{A}]. \tag{4}$$

Note that the typewriter font will be used in the remainder of this work for Choi states of a single time map (A and U), whereas a capital Greek letter will be used for the Choi state of a process ($\Upsilon$ and $\Omega$), and a capital boldfont Latin letter for a multitime instrument Choi state ($\mathbf{A_k}$), which we define below.

Expanding on the above, we can construct the Choi state of a composition of two maps $\mathcal{A} \circ \mathcal{B}$, for $\mathcal{B} : \mathcal{H}_a \to \mathcal{H}_b$ and $\mathcal{A}$ as above, via the link product [29],

$$\mathtt{A} * \mathtt{B} := \mathrm{tr}_b[(\mathbb{1}_c \otimes \mathtt{A}^{\mathrm{T}_b})(\mathtt{B} \otimes \mathbb{1}_a)], \tag{5}$$

where $\mathtt{B} \in \mathcal{L}(\mathcal{H}_b \otimes \mathcal{H}_a)$ and $\mathtt{A} \in \mathcal{L}(\mathcal{H}_c \otimes \mathcal{H}_b)$. That is, we append identity matrices to $\mathtt{A}$ and $\mathtt{B}$ such that they live on the same Hilbert space, $\mathcal{L}(\mathcal{H}_a \otimes \mathcal{H}_b \otimes \mathcal{H}_c)$, and then trace over the shared space $\mathcal{H}_b$. Here, the superscript $\mathtt{A}^{\mathrm{T}_b}$ means the partial transpose of the matrix $\mathtt{A}$ over the space $\mathcal{H}_b$. The Link product essentially describes multiplication entirely within the Choi representation, via matrix multiplication on the shared space and tensor product on the independent. This is a key tool with which we can concisely and explicitly define the process tensor.

An operator norm of instruments that will be relevant to our results is the POVM (positive operator valued measured) norm,[1] which is the largest singular value of the POVM element $\mathtt{A} := \sum_\alpha K_\alpha^\dagger K_\alpha \le \mathbb{1}$,

$$\|\mathtt{A}\|_{\mathrm{p}} := \max_{\|\psi\|_2 = 1} \|\mathtt{A}|\psi\rangle\|_2 = \max_{\sqrt{\langle\psi|\psi\rangle}=1} \sqrt{\langle\psi|\mathtt{A}^2|\psi\rangle}. \tag{6}$$

When a map is trace preserving, we have $\mathtt{A} = \sum_\alpha K_\alpha^\dagger K_\alpha = \mathbb{1}$ and the POVM norm is then equal to unity.

## 2.2 Tensor representation of quantum processes

Consider a partition of an isolated quantum system, initially in the state $\rho$, into a system ($S$) of interest and the rest, which we call an environment ($E$). $SE$ then evolves unitarily up until some point $t_1$, according to the time-independent Hamiltonian

$$H = \sum E_n P_n, \tag{7}$$

i.e., via the supermap

$$\mathcal{U}_1(\cdot) := e^{-iH\Delta t_1}(\cdot)e^{iH\Delta t_1}. \tag{8}$$

An instrument is then applied to the $S$ state at time $t_1$, described by a CP map $\mathcal{A}_1 \equiv \mathcal{A}_1 \otimes \mathbb{1}_{\mathrm{E}}$. Note that this $SE$ decomposition is consistent with the notion of a coarse (or fine) measurement on an isolated system. For example, measuring the total magnetization of a spin system is highly coarse-grained, and one can appropriately couple the system of spins ($E$) to an ancilla spin ($S$), such that measuring this ancilla will determine the total magnetization. Now, this dynamics followed by measurement is repeated, with a variable time of unitary evolution $\Delta t_i$, and choice of single time instruments $\mathcal{A}_i$, as depicted in Fig. 1 (a). The expectation value of the sequence of instruments is given by

$$\mathrm{tr}[\mathcal{A}_k \mathcal{U}_k \cdots \mathcal{A}_1 \mathcal{U}_1(\rho)] = \mathrm{tr}[\Upsilon_\mathbf{k} \mathbf{A_k}^{\mathrm{T}}] =: \langle \mathbf{A_k} \rangle_\Upsilon, \tag{9}$$

where we have introduced the process tensor $\Upsilon_\mathbf{k}$ and the multitime instrument $\mathbf{A_k}$, defined for the times $\mathbf{k} := \{t_1, t_2, \ldots, t_k\}$. Note that adjacent calligraphic characters mean the composition of maps, $\mathcal{A}_i \mathcal{U}_j := \mathcal{A}_i \circ \mathcal{U}_j$.

---

[1]We name it the POVM norm so as not to get mixed up with the operator norm of the instrument itself, that is the largest singular value of the Choi state of the instrument.

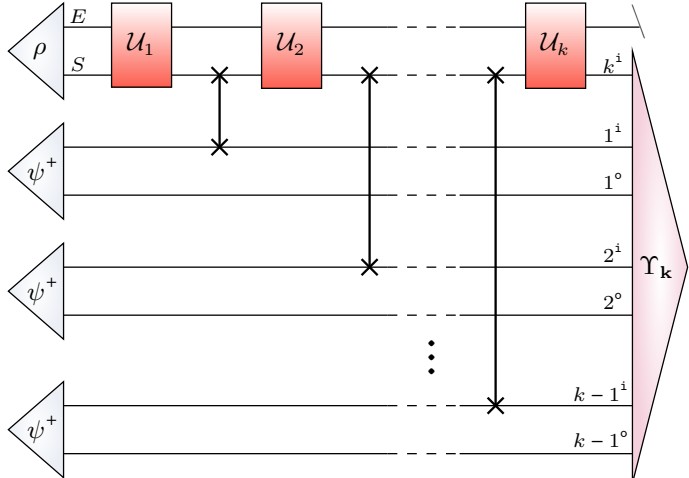

Figure 2: Circuit diagram of the construction of the Choi state of a process tensor through the generalized Choi-Jamiołkowski isomorphism [1, 28]. An ancilla system composed of $k$ (unnormalized) Bell states $\psi^+$ is appended to $SE$, with one half of a pair swapped with the $S$ state before each unitary evolution $\mathcal{U}_i$, and the $E$ space being discarded (traced over) at the end. Here the independent Hilbert spaces are labeled such that $\ell^i$ ($\ell^o$) is the input (output) index at time $t_\ell$, showing that the final tensor $\Upsilon_\mathbf{k}$ corresponds to a $2k-1$ body density matrix.

Recalling the definition of the Link product Eq. (5), we can directly convert the individual CP maps to their Choi representation, to prove Eq. (9): First, we have $\rho \in B(\mathcal{H}^o_{S,0} \otimes \mathcal{H}^o_{E,0})$, $\mathcal{A}_k : B(\mathcal{H}^i_{S,k}) \to B(\mathcal{H}^o_{S,k})$, $\mathcal{U}_{k:k-1} : B(\mathcal{H}^o_{S,k-1} \otimes \mathcal{H}^o_{E,k-1}) \to B(\mathcal{H}^i_{S,k} \otimes \mathcal{H}^i_{E,k-1})$. The notation $\mathcal{H}^{i/o}_{S(E),j}$ means the input/output Hilbert space of the system (environment) at measurement time $j$, which we give in order to specify the independent Hilbert spaces so it is clear which tensor indices contract in the following; see Fig. 2. We then write each of these in terms of their matrix indices to get the L.H.S. of Eq. (9)

$$\sum_{\text{all}} \delta_{x_3,y_3} \delta_{\alpha_3,\beta_3} \mathsf{A}_{x_3 a_3, y_3 b_3} \mathsf{U}^{\alpha_3 \alpha_2, \beta_3 \beta_2}_{a_3 x_2, b_3 y_2} \mathsf{A}_{x_2 a_2, y_2 b_2} \mathsf{U}^{\alpha_2 \alpha_1, \beta_2 \beta_1}_{a_2 x_1, b_2 y_1} \mathsf{A}_{x_1 a_1, y_1 b_1} \mathsf{U}^{\alpha_1 \alpha_0, \beta_1 \beta_0}_{a_1 x_0, b_1 y_0} \rho^{\alpha_0, \beta_0}_{x_0, y_0}. \tag{10}$$

We have used Greek indices for the environment, and Latin for the system. Upon separating all terms with Greek indices from those that only have Latin ones we have

$$\sum_{\text{latin}} \left( \sum_{\text{greek}} \mathsf{U}^{\epsilon \alpha_2, \epsilon \beta_2}_{a_3 x_2, b_3 y_2} \mathsf{U}^{\alpha_2 \alpha_1, \beta_2 \beta_1}_{a_2 x_1, b_2 y_1} \mathsf{U}^{\alpha_1 \alpha_0, \beta_1 \beta_0}_{a_1 x_0, b_1 y_0} \rho^{\alpha_0, \beta_0}_{x_0, y_0} \right) \left( \sum_e \mathsf{A}_{e a_3, e b_3} \mathsf{A}_{x_2 a_2, y_2 b_2} \mathsf{A}_{x_1 a_1, y_1 b_1} \right). \tag{11}$$

Relabelling the two terms yields Eq. (9)

$$\text{tr}[\Upsilon_\mathbf{k} \mathbf{A}^\mathsf{T}_\mathbf{k}] = \sum_{\text{latin}} (\Upsilon_\mathbf{k})_{a_3 x_2 a_2 x_1 a_1, b_3 y_2 b_2 y_1 b_1} (\mathbf{A}_\mathbf{k})_{a_3 x_2 a_2 x_1 a_1, b_3 y_2 b_2 y_1 b_1}. \tag{12}$$

While we have chosen a $k = 3$ step process as an example, a more general procedure will yield the explicit definitions

$$\begin{aligned} \Upsilon_\mathbf{k} &:= \text{tr}_E[\mathsf{U}_k * \cdots * \mathsf{U}_1 * \rho], \\ \mathbf{A}_\mathbf{k} &:= \mathsf{A}_k * \cdots * \mathsf{A}_1. \end{aligned} \tag{13}$$

An alternative circuit construction of $\Upsilon_{\mathbf{k}}$ based on the generalized Choi-Jamiołkowski isomorphism can be seen in Fig. 2. Both $\Upsilon_{\mathbf{k}}$ and $\mathbf{A_k}$ are examples of quantum combs that possess well-behaved positivity and trace properties [1, 2, 5, 29]. The former guarantees the positivity of probabilities and the latter is crucial for ensuring the causality of a process - that tracing over a final output leg of a process means the preceding input has no influence on the remainder of the process; ensuring future instruments cannot affect the statistics of past outcomes. In the derivation Eq. (10) it is assumed that the $\mathcal{A}_i$ are not correlated, in which case the Link product definition of $\mathbf{A_k}$ in Eq. (13) reduces to a tensor product. We rather allow instruments to carry quantum memory, which can be operationally realized by appending a ancilla space $W$ such that each $\mathcal{A}_i$ acts instead on the combined space $SW$, as shown in Fig. 3. In such a case $\mathbf{A_k}$ is called a tester, and is the most general way to measure a process.

For our purposes, $\Upsilon_{\mathbf{k}}$ is a universal descriptor for any multitime quantum process [2, 30] and, in particular, central for describing non-Markovian processes [1, 3, 5]. This is because all dynamics and correlations that define a process are stored in the single object $\Upsilon_{\mathbf{k}}$, which can be probed by the in-principle experimentally implementable instruments encoded in $\mathbf{A_k}$. Eq. (9) is then the multitime generalization of the Born rule [4, 31, 32], where $\Upsilon_{\mathbf{k}}$ plays the role of a state and $\mathbf{A_k}$ that of a measurement. This definition allows the computation of any temporal correlation functions on a process, accounting for the invasive nature of measurements in quantum mechanics [33].

It is worth pointing out that quantum combs arise naturally in many areas of modern quantum mechanics, including: channels with operational memory [34–37], operational quantum gravity [38–40], spatiotemporal density matrix [41], causally indefinite processes [31], quantum stochastic thermodynamics [42, 43], and the quantum-to-classical transition [8, 9]. They are, of course, central to the studies of multitime correlations in open quantum systems.

## 2.3  The diamond norm distance

In the previous section we have shown that there exists a representation for quantum processes that yields $k$-time correlations as a $(2k-1)$-body quantum state, i.e. the Choi state of the process. This allows us to define distances between two quantum processes. This is very much akin to defining distance between two probability distributions, which may represent two (classical) stochastic processes [2].

Considering two processes $\Upsilon_{\mathbf{k}}$ and $\Omega_{\mathbf{k}}$, the most natural way to compute a distance between them is to ask how well one can distinguish them using the optimal multitime measurement. The most general measurement we can perform is a tester, that is we allow the multitime instrument to carry quantum memory and so be correlated in time. We therefore define the generalized diamond norm distance [2, 29, 44] between two processes as the maximum norm difference in the expectation value of any tester on them (with normalization $1/2$),

$$D_{\blacklozenge}(\Upsilon, \Omega) := \frac{1}{2} \max_{\mathbf{A_k}} \sum_{\vec{x}} |\langle \mathbf{A}_{\vec{x}} \rangle_{\Upsilon - \Omega}|, \tag{14}$$

where we have defined the shorthand $\langle \mathbf{A}_{\vec{x}} \rangle_{\Upsilon-\Omega} := \langle \mathbf{A}_{\vec{x}} \rangle_{\Upsilon} - \langle \mathbf{A}_{\vec{x}} \rangle_{\Omega}$. This definition is motivated by the fact that a particular outcome of an instrument generates a probability distribution, $\langle \mathbf{A}_{\vec{x}} \rangle_{\Upsilon} = \mathbb{P}(x_k, x_{k-1} \ldots, x_1)$, and then Eq. (14) is simply a trace difference between probability distributions, maximized over all possible distributions that can be generated on the process $\Upsilon$. In practice, however, one does not generally have access to the optimal tester. Instead, consider a restricted set of instruments $\mathcal{M}_k = \{\mathbf{A_k}\}$ which a hypothetical experimenter has access to, that probe a process at most $k$ times. We define the operational diamond norm to be

$$D_{\mathcal{M}_k}(\Upsilon, \Omega) := \frac{1}{2} \max_{\mathbf{A_k} \in \mathcal{M}_k} \sum_{\vec{x}} |\langle \mathbf{A_k} \rangle_{\Upsilon - \Omega}|. \tag{15}$$

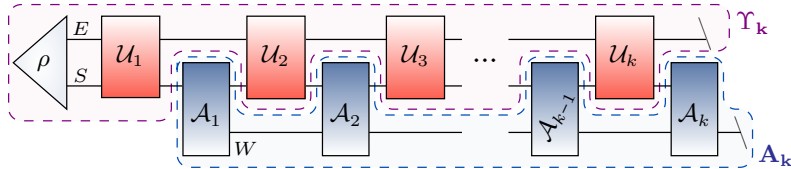

Figure 3: Allowing instruments to transmit quantum memory is equivalent to appending an ancilla space ($W$) such that $\mathbf{A_k}$ is now a quantum comb. So-called 'testers' are the most general way to probe a process $\Upsilon$ [2, 29].

In the limit of $\mathcal{M}_k$ containing all possible testers, we obtain the generalized diamond norm Eq. (14), and so $0 \leq D_{\mathcal{M}_k} \leq D_\blacklozenge \leq 1$. Intuitively, the operational diamond norm is how well one can distinguish between two processes in the best possible case, using only instruments available. This stems from the fact that we assume one has access to a limited set of measurements (for example, whatever is reasonably implementable in a given experimental setup), and measures the maximum difference in measurement statistics using the "most distinguishing" instrument available.

This is essential to our concept of process equilibration, as it allows us to describe a spatial coarse graining in the restriction of accessible instruments to a number that is operationally realistic. From now we drop the subscript $k$ on $\mathcal{M}_k$. In the single-time measurement case, where $\Upsilon \equiv \rho$ and the expectation values are the usual quantum mechanical ones, $D_\mathcal{M}$ is the analogue of the distinguishability and $D_\blacklozenge$ the trace distance [22, 45]. Note, however, that the inclusion of quantum memory in the multitime case is a non-trivial extension.

## 2.4 Underlying continuous process

So far we have considered a discrete process $\Upsilon_\mathbf{k}$, with free indices at exactly the times $\mathbf{k}$ where a chosen instrument $\mathbf{A_k}$ measures. However, in principle there exists an underlying continuous process $\Upsilon$, with an infinite set of 'free' indices at all times, with implied identity operators at each. Then, when a discrete times instrument $\mathbf{A_k}$ is chosen to measure this process, the marginal process $\Upsilon_\mathbf{k} \subset \Upsilon$ which we have derived above is used to compute the expectation value.[2] The existence of the underlying $\Upsilon$ is ensured by the Generalized Extension Theorem, the quantum generalization of the Kolmogorov Extension Theorem for classical stochastic processes [2, 30].

Considering that the underlying $\Upsilon$ can hypothetically be measured with a multitime instrument at any set of times $\mathbf{k}$, this motivates the definition of an equilibrium process, as the process generated by averaging over these times. The dynamics of such a process will be independent of times $\mathbf{k} = \{t_k, t_{k-1}, \dots, t_1\}$, with $\mathbf{k}$ specifying only the times where the process is measured. The $k$-time equilibrium process $\Omega_\mathbf{k}$ is defined as the average marginal process over all possible (ordered) sets of times $\mathbf{k}$,

$$\Omega_\mathbf{k} := \overline{\Upsilon_\mathbf{k}}^\infty = \left( \prod_{i=1}^{k} \lim_{T_i \to \infty} \frac{1}{T_i} \int_0^{T_i} d(\Delta t_i) \right) \Upsilon \tag{16}$$
$$= \mathrm{tr}_E \left[ \hat{\$} * \cdots * \hat{\$} * \rho \right],$$

where $\hat{\$}$ is the Choi state of the dephasing map with respect to the energy eigenbasis,

$$\$(\cdot) := \sum_n P_n(\cdot)P_n. \tag{17}$$

---

[2]For $\mathbf{a} := \{t_1, t_2, \dots, t_a\}$ and $\mathbf{b} := \{t_1, t_2, \dots, t_b\}$ with $a < b$, the notation $\Upsilon_\mathbf{a} \subset \Upsilon_\mathbf{b}$ means that the $a$-time process $\Upsilon_\mathbf{a}$ is related to the $b$-time process $\Upsilon_\mathbf{b}$ by insertion of identity operators at the times $\mathbf{b}\backslash\mathbf{a}$.

This means that for a given dynamic, marginal process $\Upsilon_{\mathbf{k}}$, the corresponding equilibrium process $\Omega_{\mathbf{k}}$ corresponds to replacing all global ($SE$) unitary dynamics with dephasing; see Fig. 1. We then define *process equilibration* as the time-average indistinguishability of multitime expectation values on this equilibrium process in comparison to an arbitrary non-equilibrium process $\Upsilon_{\mathbf{k}}$, via the operational diamond norm distance (15):

$$\overline{D_{\mathcal{M}_k}(\Upsilon_{\mathbf{k}}, \Omega_{\mathbf{k}})} \ll 1 \,. \tag{18}$$

This is the key definition of this work. Note that in comparison to [25], here we consider the average in Eq. (18) to be over finite time intervals.

From now on we will generally drop the subscript $\mathbf{k}$ on processes, with the understanding that the multitime instrument $\mathbf{A}_{\mathbf{k}}$ dictates the times $\mathbf{k} = (t_1, t_2, \ldots, t_k)$ at which the underlying process $\Upsilon$ marginalizes to.

## 2.5 Equilibration of processes over infinite times

The results of this work are an extension of the infinite time results from the related work [25], which we will now summarize. Specifically, the following bound was proven, in terms of the expectation value of any $k$-time instrument $\mathbf{A}_{\mathbf{k}}$ applied to a process $\Upsilon$ and its corresponding equilibrium $\Omega$,

$$\overline{|\langle \mathbf{A}_{\mathbf{k}}\rangle_{\Upsilon-\Omega}|^2}^{\infty} \leq \max_{j \in [0, k-1]} \frac{(2^k - 1)\|\mathsf{A}_{k:\cdots:(j+1)}\|_{\mathrm{p}}^2}{d_{\mathrm{eff}}[\mathcal{A}_j(\omega_j)]} \,. \tag{19}$$

Here, $\mathsf{A}_{k:\cdots:j}$ is the POVM element of the composition of CP maps

$$\mathcal{A}_k \$_k \mathcal{A}_{k-1} \cdots \$_{j+1} \mathcal{A}_j \,, \tag{20}$$

and we have defined the intermediate equilibrium state for $j \in [0, k-1] := \{0, 1, \ldots, k-1\}$,

$$\omega_j := \$_j \mathcal{A}_{j-1} \$_{j-1} \mathcal{A}_{j-2} \cdots \mathcal{A}_1 \$(\rho) \,. \tag{21}$$

The crucial term of the right hand side is the effective dimension, which is defined as

$$d_{\mathrm{eff}}[\sigma] := \frac{1}{\mathrm{tr}[\$(\sigma)^2]} \,. \tag{22}$$

In generic physical situations the effective dimension scales exponentially with system size $N$ [20, 24, 46], and is considered a quantifier for the validity of a statistical description of a many-body system. Therefore, with $\|\mathsf{A}_{k:\cdots:(j+1)}\|_{\mathrm{p}}^2 \leq 1$ acting as a scale and $k \ll N$ restricted by physical considerations, the right hand side of Eq. (19) is extremely small in typical many-body systems, leading directly to physical results on the equilibration of quantum processes over infinite time intervals, without energy gap degeneracies [25]. However, taking the infinite time average means that nothing can be said from this result about the time scales necessary to witness equilibration. It corresponds to averaging up to the recurrence time, which is typically doubly exponential in system size and so even for relatively small many body system these times can be longer than the age of the universe [47–49]. Instead, we here ask if similar equilibration results apply when averaging over finite time intervals, that are less than the recurrence times? and therefore can this give any additional insights into the open question of equilibration time scales [46, 50–55]? We will now explore this, first extending Eq. (19) to finite time intervals between instruments while allowing arbitrary energy gap degeneracies, in the spirit of Refs. [46, 50].

# 3 Equilibration of processes over finite times

Consider an isolated quantum process $\Upsilon$ as described in Section 2.2 and represented by the purple dashed comb in Fig. 1 (a). This encompasses a finite quantum system evolving according to a time-independent Hamiltonian $H = \sum E_n P_n$, which we allow to have arbitrarily degenerate energy levels. Between the global system plus environment ($SE$) unitary evolution, consider repeated, local measurements on the space $S$ alone, at the times $t_1, t_2, \ldots, t_k$ by instruments $\mathcal{A}_1, \mathcal{A}_2, \ldots, \mathcal{A}_k$. Together this represents the expectation value of a $k$-time instrument $\mathbf{A_k}$ on a process $\Upsilon$, shown in Fig. 1 (a). We then wish to investigate how the expectation value of this instrument varies when measured with respect to $\Upsilon$ in comparison to the corresponding equilibrated $\Omega$, over finite time intervals $T_\ell$ (such that $\Delta t_\ell := t_\ell - t_{\ell-1} \le T_\ell$). In contrast to the infinite time result Eq. (19), we additionally allow energy gap degeneracies and this will be included quantitatively in our results.

We will first state our main result on the equilibration of quantum processes over finite times, discuss its implications, and then detail the proof of this result in section 3.2. A reader not interested in the details of the proof should skip this section and read on to section 4, where we show how this result readily implies the equilibration of arbitrary geometric measures of quantum processes.

## 3.1 Main result

Consider the difference in expectation values of some multitime instrument $\mathbf{A_k}$, acting at the set of times $\mathbf{k} := \{t_1, t_2, \ldots, t_k\}$, between a process $\Upsilon$ and corresponding equilibrium process $\Omega$ and averaged over the time intervals $\Delta t_\ell \in [0, T_\ell]$ for each $\ell \in [1, k]$. Define finite time-averaging as,

$$\overline{X}^{\vec{T}} = \Big( \prod_{i=1}^{k} \frac{1}{T_i} \int_0^{T_i} d(\Delta t_i) \Big) X \,, \tag{23}$$

where $X$ is some function of $\Delta t_1, \Delta t_2, \ldots, \Delta t_k$. Then in full generality, any process and multitime instrument satisfy the following bound, for time intervals $T_\ell \ge 0$ and $\epsilon > 0$,

$$\overline{|\langle \mathbf{A_k} \rangle_{\Upsilon - \Omega}|^2}^{\vec{T}} \le \frac{2^{3k-1} \mathfrak{g}^k}{d_{\text{eff}}[\rho]_{\min}} =: \frac{C_k(\epsilon, T_{\min})}{d_{\text{eff}}[\rho]_{\min}} \,, \tag{24}$$

where

$$\mathfrak{g} := N(\epsilon) \Big( 1 + \frac{8 \log_2 d_{\mathrm{H}}}{\epsilon T_{\min}} \Big) \quad \text{and} \quad d_{\text{eff}}[\rho]_{\min} := \min_{\sigma \in \{\rho, \omega, \mathcal{A}_1(\rho), \mathcal{A}_1(\omega), \ldots\}} d_{\text{eff}}[\sigma] \,. \tag{25}$$

Here, $d_H$ is the number of non-degenerate energy levels, $T_{\min} := \min_{\ell \in [1,k]} T_\ell$ and, following Ref. [50], $N(\epsilon)$ is the maximum number of energy gaps in an interval of size $\epsilon > 0$,

$$N(\epsilon) := \max_E |\{(m, n) : E \le E_m - E_n \le E + \epsilon\}| \,. \tag{26}$$

When $\epsilon \to 0^+$, this reduces to the maximum degeneracy of any single energy gap, $D_G$. Eq. (24) is the finite time generalization of Eq. (19) and the main mathematical result from which our physical results are derived.

The key feature of the bound Eq. (24) is that it scales with the smallest effective dimension at any stage of either of the processes $\Upsilon$ and $\Omega$. This means that the right hand side will typically scale exponentially with system size (when there are many significantly interacting energy eigenstates), as long as no instrument 'knocks' the total $SE$ state into a small energy subspace. Also note that while $\mathfrak{g} \ge 1$ can be very large in general, in the bound (24) it scales

logarithmically with total dimension, whereas the effective dimension typically scales linearly with $d_H$ in a many body system. It will be discussed below what time scales are needed for the right hand side of the bound to be small. This will follow argumentation of Ref. [50].

Using this bound, we arrive at our main physical result on the distinguishability of the processes $\Upsilon$ and $\Omega$.

**Result 1.** *For any quantum process $\Upsilon$ with corresponding equilibrated $\Omega$, and given a set of multitime measurements $\mathcal{M}$, then for $T_\ell > 0$, $\epsilon > 0$ where $\ell \in [1, k]$,*

$$\overline{D_{\mathcal{M}}(\Upsilon, \Omega)}^{\vec{T}} \leq \frac{\mathcal{S}_{\mathcal{M}} \sqrt{C_k(\epsilon, T_{\min})}}{2 \sqrt{d_{\text{eff}}[\rho]_{\min}}}. \tag{27}$$

*Here, $\mathcal{S}_{\mathcal{M}}$ is the total combined number of outcomes for all instruments in the set $\mathcal{M}$,*

$$\mathcal{S}_{\mathcal{M}} = \sum_{M \in \mathcal{M}} \text{card}(M), \tag{28}$$

*where* $\text{card}(M)$ *is the cardinality of the set $M$.*

*Proof.* A proof for this applies Eq. (24) to a result from Ref. [25], and is given in App. D. $\square$

Assuming that process equilibration occurs in the infinite time intervals case [25], i.e. assuming that $\mathcal{S}_{\mathcal{M}} \sqrt{2^k - 1} \ll \sqrt{d_{\text{eff}}[\rho]_{\min}}$, we additionally have equilibration within finite time intervals $\vec{T}$ given that

$$T_\ell \gtrsim \frac{\log_2 d_H}{\epsilon} \qquad \text{and} \qquad k \ll \frac{1}{3} \log_2 d_{\text{eff}}[\rho]_{\min}. \tag{29}$$

The first condition states that if no energy gap is hugely degenerate and each time interval $T_\ell$ is big enough, the factor $\mathfrak{g}$ is not too large. Physically this time scale is much smaller than the recurrence times, which typically scale exponentially with effective dimension (doubly exponentially with system size). However, in realistic physical examples $\mathfrak{g}$ can blow up due to the $N(\epsilon)$ factor defined in Eq. (26). We therefore expect tighter bounds than Eq. (24) to hold with additional physical assumptions, with strong evidence that realistic (locally interacting) models with generic initial states equilibrate significantly faster than our bound (and the bound of Ref. [50]) suggests [13, 14, 56, 57]. Although it should be noted there exists relevant physical examples of transitionally invariant lattice models where it takes exponentially long to equilibrate in the single time sense [24, 58]. The strength of the present work is that it holds analytically under minimal assumptions, where analogous results are difficult to obtain with increasing physical assumptions (although some progress in the multitime case has been made very recently, see Refs. [26, 59]).

The second condition ensures the number of times at which an instrument measures the system is far less than the number of components in the system. Clearly this is satisfied in typical many body situations, with for example $\mathcal{O}(10^{23})$ particles, where measuring even $\mathcal{O}(10^2)$ time correlations is experimentally unfeasible. Physically, choosing a (small) finite $k$ is a kind of coarse graining in time; analogous to the assumption that $\mathcal{S}_{\mathcal{M}}$ is small, which is a coarse graining in (Hilbert) space. This is essential in defining process equilibration. As a counter example, a perfectly fine grained instrument which continuously measures a process at all times can distinguish a dynamical out-of-equilibrium $\Upsilon$ from a stationary $\Omega$.

A disadvantage of this result in comparison to the infinite time one, is that it cannot easily be interpreted in terms of a probability bound, such as Chebyshev's inequality [25]. This is because the bound Eq. (24) (the variance) is computed over a finite times uniform distribution, whereas the equilibrated process expectation value $\langle \mathbf{A_k} \rangle_\Omega$ (the mean) is computed over an infinite times one. Nonetheless, it offers additional insight when interpreted via the operational

diamond norm in Eq. (15), allowing us to show quantum process equilibration in finite time intervals that are generally much less than recurrence times.

Under the conditions (29), Result 1 indicates that processes equilibrate within finite time intervals. However, due to the generality of the setup, the time scales involved are still typically very large. Additional assumptions on the physical scenario would be needed for an estimate on realistic equilibration times. To this purpose, the process tensor formalism directly allows for physical assumptions on the dynamical details of a system. This will be immediately clear from a physical corollary to Result 1, which we describe in Section 4.

### 3.2 Proof of Eq. (24)

We will here derive our main result explicitly for $k = 2$ instruments, and then motivate a generalization to arbitrary $k$, with further details set out in the Appendix. We wish to bound $\overline{|\langle \mathbf{A_k} \rangle_\Upsilon - \langle \mathbf{A_k} \rangle_\Omega|^2}^{\vec{T}}$, where

$$\overline{X}^{\vec{T}} := \frac{1}{T_1 \cdots T_k} \int_0^{T_k} \cdots \int_0^{T_1} X \, d(\Delta t_1) \cdots d(\Delta t_k) \tag{30}$$

is the finite-time average over all evolution time intervals $\Delta t_i$ within the range $[0, T_i]$.

Recalling the multitime Born rule Eq. (9), we can expand unitary evolution operators in the energy eigenbasis for the following,

$$\langle \mathbf{A_k} \rangle_\Upsilon - \langle \mathbf{A_k} \rangle_\Omega = \sum_{n_i, m_i} \mathrm{tr}[\mathcal{A}_k \mathcal{P}_{n_k m_k} \cdots \mathcal{A}_1 \mathcal{P}_{n_1 m_1}(\rho)] \left\{ \prod_{i=1}^k e^{-i\Delta t_i (E_{n_i} - E_{m_i})} - \prod_{i=1}^k \delta_{m_i n_i} \right\}, \tag{31}$$

where $\mathcal{P}_{n_i m_j}(\cdot) := P_{n_i}(\cdot) P_{m_j}$ is the superoperator that projects onto the $(n_i, m_j)$th component in the energy eigenbasis.

Specifying to $k = 2$ we have,

$$\langle \mathbf{A_k} \rangle_{\Upsilon - \Omega} \underset{(k=2)}{=} \overbrace{\sum_{\substack{n_1 \neq m_1 \\ n_2 \neq m_2}} \mathrm{tr}[\mathcal{A}_2 \mathcal{P}_{n_2 m_2} \mathcal{A}_1 \mathcal{P}_{n_1 m_1}(\rho)] \left[ e^{-i\Delta t_2 (E_{n_2} - E_{m_2})} e^{-i\Delta t_1 (E_{n_1} - E_{m_1})} \right]}^{=:X}$$

$$+ \underbrace{\sum_{n_1 \neq m_1} \mathrm{tr}[\mathcal{A}_2 \$_2 \mathcal{A}_1 \mathcal{P}_{n_1 m_1}(\rho)] e^{-i\Delta t_1 (E_{n_1} - E_{m_1})}}_{=:Y} + \underbrace{\sum_{n_2 \neq m_2} \mathrm{tr}[\mathcal{A}_2 \mathcal{P}_{n_2 m_2} \mathcal{A}_1(\omega_1)] e^{-i\Delta t_2 (E_{n_2} - E_{m_2})}}_{=:Z}. \tag{32}$$

We will now multiply this with its complex conjugate and independently time-average over each $\Delta t_\ell$ over the range $[0, T_\ell]$, in order to obtain the desired quantity $\overline{|\langle \mathbf{A_k} \rangle_\Upsilon - \langle \mathbf{A_k} \rangle_\Omega|^2}^{\vec{T}}$. We label the indices corresponding to complex conjugate parts with primes, $n_i'$ and $m_i'$, and after taking the modulus square we obtain exponential multiplicative factors with exponents $-i\Delta t_\ell (E_{n_\ell} - E_{n_\ell'} - E_{m_\ell} + E_{m_\ell'})$ and $-i\Delta t_\ell (E_{n_\ell} - E_{m_\ell})$. These factors contain all the time dependencies, and so we therefore define the tensors $\mathcal{G}^{(\ell)}$ and $G^{(\ell)}$ with components

$$\overline{\mathcal{G}}^{(\ell)}_{n_\ell m_\ell n_\ell' m_\ell'} := \overline{\exp[-i\Delta t (E_{n_\ell} - E_{n_\ell'} - E_{m_\ell} + E_{m_\ell'})]}^{T_\ell},$$
$$\overline{G}^{(\ell)}_{n_\ell m_\ell} := \overline{\exp[-i\Delta t (E_{n_\ell} - E_{m_\ell})]}^{T_\ell} \tag{33}$$

in the corresponding energy eigenbasis. For brevity we will also define the following multilinear function

$$f(\mathcal{X}, \mathcal{Y}) := \mathrm{tr}[\mathcal{A}_2 \mathcal{X} \mathcal{A}_1 \mathcal{Y}(\rho)]. \tag{34}$$

So for example $f(\mathcal{P}_{n_2 m_2}, \mathcal{P}_{n_1 m_1})$, $f(\$, \mathcal{P}_{n_1 m_1})$, and $f(\mathcal{P}_{n_2 m_2}, \$)$ appear in Eq. (32) .

Then for $k = 2$, we obtain

$$\overline{|\langle \mathbf{A_k}\rangle_{\Upsilon - \Omega}|^2}^{\vec{T}} \underset{(k=2)}{=} \overline{X^2} + \overline{Y^2} + \overline{Z^2} + 2\text{Re}\{\overline{XY^*} + \overline{XZ^*} + \overline{YZ^*}\}. \tag{35}$$

We will address each of these terms in turn, and see that they are all bounded by a quantity that scales with an effective dimension, using results from Refs. [25, 50]. There are three different type of terms in Eq. (35): the first three only have double sums over $n_\ell^{(\prime)} \neq m_\ell^{(\prime)}$ (where a prime with a bracket here means a sum over both prime and non-prime indices). We call these 'diagonal'; the next two terms have both a double sum over $n_\ell^{(\prime)} \neq m_\ell^{(\prime)}$ and an independent sum over $n_j \neq m_j$ with $j \neq \ell$. We call these 'cross terms'; and the final term contains only independent sums over $n_\ell \neq m_\ell$, which we call 'off-diagonal'. These three type of terms require somewhat different methods to bound, but classify all the different terms that appear at higher $k$. The following methods of will therefore directly generalize to many time instruments.

### 3.2.1 Diagonal terms

Looking at the first term, we have

$$\overline{X^2} = \sum_{\substack{n_1^{(\prime)} \neq m_1^{(\prime)} \\ n_2^{(\prime)} \neq m_2^{(\prime)}}} \overline{\mathcal{G}}^{(1)}_{n_1 m_1 n_1' m_1'} \overline{\mathcal{G}}^{(2)}_{n_2 m_2 n_2' m_2'} f(\mathcal{P}_{n_2 m_2}, \mathcal{P}_{n_1 m_1}) f(\mathcal{P}_{n_2' m_2'}, \mathcal{P}_{n_1' m_1'})^*, \tag{36}$$

where $\sum_{n_\ell^{(\prime)} \neq m_\ell^{(\prime)}} := \sum_{n_\ell \neq m_\ell} \sum_{n_\ell' \neq m_\ell'}$ (with $\ell \in \{1, 2\}$ here). The indices with primes, ($n_i'$ and $m_i'$), come from the complex conjugate $X^*$, and the complex prefactor $\mathcal{G}^{(i)}_{n_i m_i n_i' m_i'}$ (a tensor with four indices) is defined in Eq. (33). We can write this prefactor as a matrix, by gathering the indices as $\alpha \equiv (n_\ell, m_\ell)$. One can then see that $\overline{\mathcal{G}}^{(\ell)}_{n_\ell m_\ell n_\ell' m_\ell'} \equiv \overline{\mathcal{G}}^{(\ell)}_{\alpha \alpha'}$ is Hermitian in $\alpha$, and similarly $M_{\alpha \alpha'} := \overline{\mathcal{G}}^{(1)}_{n_1 m_1 n_1' m_1'} \overline{\mathcal{G}}^{(2)}_{n_2 m_2 n_2' m_2'}$ is Hermitian in the indices $\alpha = (n_1, m_1, n_2, m_2)$. We may therefore use that for Hermitian $M$, $\sum_{\alpha \alpha'} v_\alpha^* M_{\alpha \alpha'} v_{\alpha'} \leq \|M\| \sum_\alpha |v_\alpha|^2$, where $\|M\|$ is the usual operator norm of the matrix $M$.[3] Therefore,

$$\overline{X^2} \leq \|\overline{\mathcal{G}}^{(1)}\| \|\overline{\mathcal{G}}^{(2)}\| \sum_{\substack{n_1 \neq m_1 \\ n_2 \neq m_2}} |f(\mathcal{P}_{n_2 m_2}, \mathcal{P}_{n_1 m_1})|^2. \tag{37}$$

From here, we will use an identity used to obtain the infinite time result of Ref. [25],

$$\sum_{\substack{n_i \neq m_i \\ \cdots \\ n_j \neq m_j}} \left| \text{tr}[\mathcal{A}_k \mathcal{D}_k \ldots \mathcal{D}_{j+1} \mathcal{A}_j \mathcal{P}_{n_j, m_j} \mathcal{A}_{j-1} \mathcal{S}_{j-1} \ldots \mathcal{A}_j \mathcal{S}_1(\rho)] \right| \leq \|\mathsf{A}_{k:\cdots:j}\|_p^2 d_{\text{eff}}^{-1}[\mathcal{A}_{j-1}(\omega_{j-1})], \tag{38}$$

where $1 \leq i < j \leq k$, $\mathcal{D}_\ell \in \{\$, \mathcal{I}\}$ for identity superoperator $\mathcal{I}$, and $\mathcal{S}_\ell \in \{\$, \mathcal{P}_{n_\ell m_\ell}\}$. The norm $\|\mathsf{A}_{k:\cdots:j}\|_p^2$ is the POVM norm of the composition of CP maps $\mathcal{A}_k \mathcal{D}_k \mathcal{A}_{k-1} \cdots \mathcal{D}_{j+1} \mathcal{A}_j$, as defined in Eq. (6). The key thing to note is that the choice of each $\mathcal{S}_\ell$ does not matter for the inequality (38), instead only the final (leftmost) projector is what determines the bound. A proof of this is given in App. A for completeness, reproduced from Ref. [25]. We then obtain

$$\overline{X^2} \leq g_1 g_2 \|\mathsf{A}_2\|_p^2 d_{\text{eff}}^{-1}[\mathcal{A}_1(\omega_1)], \tag{39}$$

---

[3]The operator norm of a matrix is its largest singular value; formally, $\|M\| := \sup\{\|Mv\| : \|v\| = 1\}$.

where we have also introduced a bound derived in Ref. [50],

$$\|\overline{\mathcal{G}}^{(\ell)}\| \leq N(\epsilon)\left(1 + \frac{8\log_2 d_{\mathrm{H}}}{\epsilon T_\ell}\right) =: g_\ell\,, \tag{40}$$

where $\epsilon > 0$, $d_{\mathrm{H}}$ is the number of distinct energies, and $N(\epsilon)$ is maximum number of energy gaps in an interval of size $\epsilon$, as defined in Eq. (26). A proof for Eq. (40) is given in Appendix B for completeness. An equivalent method can be applied to bound the other two diagonal terms,

$$\overline{Y^2} \leq g_1\|A_{2:1}\|_{\mathrm{p}}^2 d_{\mathrm{eff}}^{-1}[\rho] \quad \text{and} \quad \overline{Z^2} \leq g_2\|A_2\|_{\mathrm{p}}^2 d_{\mathrm{eff}}^{-1}[\mathcal{A}_1(\omega_1)]\,. \tag{41}$$

### 3.2.2 Cross terms

Now, the next two cross terms of Eq. (35) will also be proportional to an effective dimension, but require a different treatment and obtain an additional multiplicity.

$$2\mathrm{Re}\{\overline{XY^*}\} = 2\mathrm{Re}\left\{\sum_{\substack{n_1^{(\prime)} \neq m_1^{(\prime)} \\ n_2 \neq m_2}} \overline{\mathcal{G}}^{(1)}_{n_1 m_1 n_1' m_1'} \overline{G}^{(2)}_{n_2 m_2} f(\mathcal{P}_{n_2 m_2}, \mathcal{P}_{n_1 m_1}) f(\$, \mathcal{P}_{n_1' m_1'})^*\right\}$$

$$\leq 2\overline{G}^{(2)}_{\max}\left|\sum_{n_1^{(\prime)} \neq m_1^{(\prime)}} \overline{\mathcal{G}}^{(1)}_{n_1 m_1 n_1' m_1'} f\left(\sum_{n_2 \neq m_2} \mathcal{P}_{n_2 m_2}, \mathcal{P}_{n_1 m_1}\right) f(\$, \mathcal{P}_{m_1' n_1'})\right|\,, \tag{42}$$

where we have used that for $z \in \mathbb{C}$, $\mathrm{Re}(z) \leq |z|$, and defined the max value of the matrix $\overline{G}^{(\ell)}$, $G^{(\ell)}_{\max} := \max_{m \neq n}\left|G^{(\ell)}_{nm}\right| \leq 1$. Note also that the complex conjugate of the trace function $f$ (Eq. (34)) corresponds only to a transpose of indices $m$ and $n$. Noticing that $\sum_{n \neq m} \mathcal{P}_{nm} \equiv \mathcal{I} - \$$, we can use the linearity of $f$ and the triangle inequality to expand the sum over $n_2 \neq m_2$. For the sum over $n_1^{(\prime)} \neq m_1^{(\prime)}$, we may again use that $\mathcal{G}^{(\ell)}_{nmn'm'}$ is Hermitian, and so $\sum_{\alpha\alpha'} u_\alpha M_{\alpha\alpha'} v_{\alpha'}$ defines an inner product. Applying the Cauchy-Schwarz inequality with respect to this, after the triangle inequality mentioned above, we arrive at

$$2\mathrm{Re}\{\overline{XY^*}\} \leq 2\overline{G}^{(2)}_{\max}\|\overline{\mathcal{G}}^{(1)}\|\sqrt{\sum_{n_1' \neq m_1'}|f(\$, \mathcal{P}_{n_1' m_1'})|^2}\left(\sqrt{\sum_{n_1 \neq m_1}|f(\mathcal{I}, \mathcal{P}_{n_1 m_1})|^2} + \sqrt{\sum_{n_1 \neq m_1}|f(\$, \mathcal{P}_{n_1 m_1})|^2}\right). \tag{43}$$

At this point we are left with terms of the form of the identity Eq. (38), and so arrive at the final bound for this term,

$$2\mathrm{Re}\{\overline{XY^*}\} \leq 4s_2\,g_1\|A_{2:1}\|_{\mathrm{p}}^2 d_{\mathrm{eff}}^{-1}[\rho]\,, \tag{44}$$

where $\Delta E_{\min}$ is the minimum energy gap, and we have computed

$$\begin{aligned}G^{(\ell)}_{\max} &= \max_{m \neq n}\frac{2|\sin[T_\ell(E_m - E_n)/2]|}{T_\ell|E_m - E_n|} \\ &= |\mathrm{sinc}\left(\frac{T_\ell \Delta E_{\min}}{2}\right)| =: s_\ell \leq 1\,.\end{aligned} \tag{45}$$

The inequality is saturated if and only if the time interval $T_\ell$ is finite and the Hamiltonian $H$ has a degenerate energy level. Note however, that we may in full generality consider only non-degenerate Hamiltonians as this generalizes to degenerate Hamiltonian's via an additional

inequality in convex mixtures of pure instruments and pure initial states.[4] Additionally considering that we here only consider finite $T_\ell$, the inequality in Eq. (45) is strict.

The fifth term of Eq. (35) can be upper bound in a similar manner,

$$2\text{Re}\{\overline{XZ^*}\} \le 2\overline{G}_{\text{max}}^{(1)}\|\overline{\mathcal{G}}^{(2)}\|\|A_2\|_p^2\Big(\sqrt{d_{\text{eff}}^{-1}[\mathcal{A}_1(\omega_1)]}\sqrt{d_{\text{eff}}^{-1}[\mathcal{A}_1(\rho)]} + d_{\text{eff}}^{-1}[\mathcal{A}_1(\omega_1)]\Big) \tag{46}$$

$$\le 4s_1\,g_2\,\|A_2\|_p^2\,d_{\text{eff}}^{-1}[\rho]_{\text{min}}, \tag{47}$$

where we have used the minimum effective dimension at any stage of either process, as defined in Eq. (25).

### 3.2.3 Off-diagonal terms

We are finally left to bound the last term of Eq. (35), which is 'off-diagonal in the sense that it contains no double-sums over $n_\ell^{(\prime)} \ne m_\ell^{(\prime)}$. The time averages for $Y$ and $Z^*$ are entirely independent, and so

$$\begin{aligned}
2\text{Re}\{\overline{YZ^*}\} &\le 2|\overline{f(\$,\mathcal{U}_1-\$)}^{T_1}\overline{f(\mathcal{U}_2-\$,\$)}^{T_2}| \\
&\le \sqrt{\overline{f(\$,\mathcal{U}_1-\$)^2}^{T_1}\overline{f(\mathcal{U}_2-\$,\$)^2}^{T_2}} \\
&\le \sqrt{\frac{\|\overline{\mathcal{G}}^{(1)}\|\|A_{2:1}\|_p^2}{d_{\text{eff}}[\rho]}\frac{\|\overline{\mathcal{G}}^{(2)}\|\|A_2\|_p^2}{d_{\text{eff}}[\mathcal{A}_1(\omega)]}} \\
&\le \frac{\sqrt{g_1\|A_{2:1}\|_p^2\,g_2\|A_2\|_p^2}}{d_{\text{eff}}[\rho]_{\text{min}}}.
\end{aligned} \tag{48}$$

Here we have used the single time equilibration result of Ref. [50], which can be derived using the identity (38) together with Eq. (40).

### 3.2.4 Final Bound for $k = 2$

We can now combine Eqs. (39), (41), (44), (46), and (48) to obtain the full equilibration bound for $k = 2$,

$$\begin{aligned}
\overline{|\langle \mathbf{A_k}\rangle_{\Upsilon-\Omega}|^2}^{\vec{T}} \le \Big(&g_1g_2\|A_2\|_p^2 + g_1\|A_{2:1}\|_p^2 + g_2\|A_2\|_p^2 \\
&+ 4g_1s_2\|A_{2:1}\|_p^2 + 4g_2s_1\|A_2\|_p^2 \\
&+ 2\sqrt{g_1g_2\|A_2\|_p^2\|A_{2:1}\|_p^2}\Big)d_{\text{eff}}^{-1}[\rho]_{\text{min}},
\end{aligned} \tag{49}$$

where we have introduced an additional inequality to get the common factor $d_{\text{eff}}^{-1}[\rho]_{\text{min}}$ (which is defined in Eq. (25)).

A perceptive reader may notice that Eq. (49) does not exactly reduce to the infinite time bound Eq. (19). As when $T_\ell \to \infty$, then $g_\ell \to 1$ and $s_\ell \to 0$, and so Eq. (49) reduces to,

$$\overline{|\langle \mathbf{A_k}\rangle_{\Upsilon-\Omega}|^2}^{\infty} \le \Big(\|A_2\|_p^2 + \|A_{2:1}\|_p^2 + \|A_2\|_p^2 + 2\sqrt{\|A_2\|_p^2\|A_{2:1}\|_p^2}\Big)d_{\text{eff}}^{-1}[\rho]_{\text{min}}, \tag{50}$$

---

[4]Via a similar argument to one given in Ref. [22], we can ensure that the evolution is equivalently according to a non-degenerate Hamiltonian by choosing a basis for any degenerate subspace such that the $SE$ state at any particular time step may overlap only with one of the degenerate basis states for each distinct energy. This argument holds for pure states at all times, and so only for purity-preserving instruments $\mathcal{A}_i$. However, $\overline{|\langle \mathbf{A}\rangle_{\Upsilon-\Omega}|^2}^{\infty}$, is convex in mixtures of pure instruments/states, so any bound for pure instruments directly implies a bound for mixed instruments/states.

which is a slightly looser bound than Eq. (19). This comes from the fact that the final term which we bounded, $2\mathrm{Re}\{\overline{YZ^*}\}$, can in fact alternatively be bounded by a term $\propto s_1 s_2$, without any inverse effective dimension dependence. This then goes to zero in the infinite time limit. However, in the finite time case this term can be problematic: $\Delta E_{\min}$ can be extremely small in a typical many body system (with a large $d_{\mathrm{eff}}[\rho]$), and so this term could cause the bound to become large in such a case, predicting larger equilibration times for systems with smaller $\Delta E_{\min}$. In summary, the final term of Eq. (49) can be replaced with,

$$2\mathrm{Re}\{\overline{YZ^*}\} \leq \quad \min\left\{2\sqrt{g_1 g_2 \|\mathsf{A}_2\|_{\mathrm{p}}^2 \|\mathsf{A}_{2:1}\|_{\mathrm{p}}^2 d_{\mathrm{eff}}^{-1}[\rho]_{\min}}, 8s_1 s_2\right\}, \tag{51}$$

in which case we arrive at the infinite time limit of Ref. [25]. We will omit this for clarity, as the bound proportional to the effective dimension is the most meaningful in the case of finite time intervals.

### 3.2.5 Extension to arbitrary $k$

The derivation of the $k = 2$ result Eq. (49) dealt with the three types of terms that appear in the generalization of Eq. (35) for arbitrary $k$-time instruments. Therefore, the methods here can be directly extended to a similar derivation for a bound for any $k$. We give the result explicitly for $k = 3$ in App. C (49 terms). The key factor is the inverse proportionality with respect to the effective dimension, which will typically be large compared to the other multiplicative factors, and compared to the total multiplicity.

Finally, using that $s_\ell = \mathrm{sinc}(\Delta E_{\max} T_\ell/2) \leq 1$, $g_\ell \geq 1$, and that $\|\mathsf{A}\| \leq 1$, we can generalize the above derivation to arrive at Eq. (24), together with a multiplicity argument which can be found in App. C. This completes the proof.

## 4 Equilibration of geometric measures

The multitime correlation equilibration results of Eqs. (19) and (24) are stronger than previous equilibration results for singletime observables [20–22, 46, 50]. To see this, in this section we show how process equilibration directly implies the equilibration of a number of geometric measures describing physical properties of a quantum process. We first provide a general theorem on the equilibration of geometric measures, and then specify some examples of physical measures. A geometric measure $\mathcal{E}_{\mathcal{M}}$ is defined as the minimum distance to the closest process $\Lambda \in \mathcal{K}$, where $\mathcal{K}$ is some restricted set of processes that defines the measure,

$$\mathcal{E}_{\mathcal{M}}(\Upsilon) := \min_{\Lambda \in \mathcal{K}}\big(D_{\mathcal{M}}(\Upsilon, \Lambda)\big). \tag{52}$$

Our chosen distance metric here is the operational diamond norm under the restricted set of instruments $\mathcal{M}$, as defined in Eq. (15). This allows us to derive results about the equilibration of various geometric measures.

**Result 2.** *For any geometric measure $\mathcal{E}_{\mathcal{M}}$ of processes, in terms of the operational diamond norm distance restricted to the set of at most $k$ time instruments $\mathcal{M}$,*

$$\overline{|\mathcal{E}_{\mathcal{M}}(\Upsilon) - \mathcal{E}_{\mathcal{M}}(\Omega)|}^{\vec{T}} \leq \frac{\mathcal{S}_{\mathcal{M}}\sqrt{C_k(\epsilon, T_{\min})}}{2\sqrt{d_{\mathrm{eff}}[\rho]_{\min}}}. \tag{53}$$

*Proof.* Consider without loss of generality that $\mathcal{E}_{\mathcal{M}}(\Upsilon) \geq \mathcal{E}_{\mathcal{M}}(\Omega)$; an equivalent argument

applies in the complementary case. Then,

$$
\begin{aligned}
|\Delta\mathcal{E}_{\mathcal{M}}| &= |\mathcal{E}_{\mathcal{M}}(\Upsilon) - \mathcal{E}_{\mathcal{M}}(\Omega)| \\
&= \mathcal{E}_{\mathcal{M}}(\Upsilon) - \mathcal{E}_{\mathcal{M}}(\Omega) \\
&= \min_{\Lambda_{\Upsilon} \in \mathcal{K}} D_{\mathcal{M}}(\Upsilon, \Lambda_{\Upsilon}) - \mathcal{E}_{\mathcal{M}}(\Omega) \\
&\leq D_{\mathcal{M}}(\Upsilon, \Lambda'_{\Omega}) - \mathcal{E}_{\mathcal{M}}(\Omega) \\
&= D_{\mathcal{M}}(\Upsilon, \Lambda'_{\Omega}) - D_{\mathcal{M}}(\Omega, \Lambda'_{\Omega}),
\end{aligned}
\tag{54}
$$

where $\Lambda'_{\Omega}$ is chosen to be the process that minimizes $D_{\mathcal{M}}(\Omega, \Lambda_{\Omega})$, i.e. that in the definition of $\mathcal{E}_{\mathcal{M}}(\Omega)$. To arrive at the penultimate line, we have used that $\mathcal{E}_{\mathcal{M}}(\Upsilon)$ is a minimum over all processes in the restricted set $\mathcal{K}$, and so $\Lambda'_{\Omega}$ satisfies the inequality for the first term. Now we apply the triangle inequality to arrive at,

$$
|\Delta\mathcal{E}_{\mathcal{M}}| \leq D_{\mathcal{M}}(\Upsilon, \Omega).
\tag{55}
$$

We can then take the multitime average and apply the bound Eq. (27). $\qquad\square$

This allows us to prove that (geometric) time-dependent dynamical properties of a process equilibrate to a time-independent quantity in finite time intervals. All these bounds will result in meaningful equilibration under the same conditions as when the right hand side of Eq. (27) is small. That is, for a large effective dimension, a realistic number of total outcomes of measurements $\mathcal{S}(\mathcal{M})$ that act at a not too large number of times $k$, and given that no energy gap is hugely degenerate, the quantity $\mathcal{E}(\Upsilon)$ is approximately equal to the time-independent $\mathcal{E}(\Omega)$.

Examples of such geometric measures include the non-Markovianity [2, 3, 5, 60, 61], the entanglement in time (including genuinely multipartite entanglement) [7], and the classicality of a process [8, 9]. Given that the implications of process equilibration on non-Markovianity was investigated in Ref. [25], we will here focus on classicality.

A classical stochastic process is a joint probability distribution on a multi-time random variable, $\mathbb{P}(x_k, \ldots, x_1)$. The process tensor is the quantum generalization of this, reducing to it in the correct limit [2, 8, 9], preserving the causal order of a process and satisfying a *generalized Kolmogorov extension theorem* (GET), in that one can marginalize over time steps through the insertion of the identity super-operator $\mathcal{I}$ for an instrument, and so show the existence of an underlying process on all times [30].

A quantum process is deemed classical when it satisfies the Kolmogorov consistency condition inherent to classical stochastic processes [8, 9],

$$
\mathbb{P}(x_1, \ldots, \cancel{x_i}, \ldots, x_k) := \sum_i \mathbb{P}(x_1, \ldots, x_i, \ldots, x_k),
\tag{56}
$$

for all $i$. This means that ignoring a step of the process is equivalent to summing over all outcomes. Recalling the definition of single time instruments as CP maps on some space $S$ (see section 2.1), one can expand them in terms of projectors onto their eigenspace, $\mathcal{A}_i(\cdot) \equiv \sum_{x_i} x_i \mathcal{P}_{x_i}(\cdot)$, where $x_i$ is an outcome of the measurement. Then, in terms of the process tensor, the classical condition Eq. (56) means that if

$$
\langle \mathrm{P}_{x_k} \otimes \ldots \Delta_i \ldots \otimes \mathrm{P}_{x_1} \rangle_{\Upsilon} = \langle \mathrm{P}_{x_k} \otimes \ldots \mathbb{1} \ldots \otimes \mathrm{P}_{x_1} \rangle_{\Upsilon},
\tag{57}
$$

then the quantum process $\Upsilon$ is classical. Here, for the projector superoperator $\mathcal{P}_{x_i}$, we have called its Choi state $\mathrm{P}_{x_i}$, and defined the dephasing operation $\Delta_i := \sum_{x_i} \mathcal{P}_{x_i}(\cdot)$. That is, $\Upsilon$ is classical when marginalizing in the quantum sense (insertion of $\mathcal{I}$) is equivalent to marginalizing in the classical sense (tracing over all outcomes). To ground ourselves here among the

formalism, consider the following explicit example of an experimental realization of Eq. (57). Take an electron traversing a sequence of Stern-Gerlach apparatuses, within a noisy environment such as outside photons interacting with the electron. Then the electron corresponds to the system which we can measure, the noisy photons correspond to the environment, and the outcomes $x_j$ are whether the electron is measured as spin up or spin down according to the $j^{\text{th}}$ apparatus. Then Eq. (57) corresponds to whether the experimenter leaves the $i^{\text{th}}$ device in the sequence, allowing both spin up and spin down to continue on (left hand side), compared to removing the $i^{\text{th}}$ device altogether (right hand side). Then we call a process classical if this has no effects on the statistics of the measurements on devices $1, \ldots, i-1, i+1, \ldots, k$.

We can now define a geometric measure of the *classicality* $\mathcal{C}_{\mathcal{M}}$ of a process, as the distance to the closest classical process, given the restricted set $\mathcal{M}$ with which the process can be probed. Therefore choosing $\mathcal{E} \equiv \mathcal{C}$ in Eq. (53), we arrive at the equilibration of the classicality of a process to the time-independent equilibrium quantity $\mathcal{C}_{\mathcal{M}}(\Omega)$. This does not mean that the equilibrium process $\Omega$ is necessarily classical. However, it does say that for coarse multi-time observables and large enough typical systems, how classical the statistics of your measurements look is overwhelmingly likely to be close to this constant. This means that, solely within a (generalized) Born rule quantum measurement framework, it is extremely likely that the classicality of your measurement statistics are close to some value $\mathcal{C}_{\mathcal{M}}(\Omega)$. Note that no semiclassical limit is taken here, and so this is a step towards a quantum process description of the emergence of classical stochasticity. If there is extra structure on the quantum process, such as an assumption of quantum Darwinism [62, 63], this could have profound implications for the emergence of macroscopic, objective determinism.

Our ultimate goal is to find the constraints that lead to nontrivial dynamical phenomena for a system, i.e., non-Markovian processes, from an underlying system-environment unitary quantum process. This has the potential to bridge the gaps between the quantum and classical theories.

# 5 Conclusions and discussion

In this work we have proven the conditions under which equilibration of quantum processes occurs in finite time intervals. This is a generalization of the infinite time process equilibration results of Ref. [25], and the extension of the finite time equilibration results of Refs. [46, 50] to multitime observables.

The time scales involved are generally much less than the recurrence times, and so this work offers a method on approximating equilibration times based on the properties of quantum processes. An example of a possible application of the bounds on geometric measures (Result 2) would be to enforce that $\mathcal{E}_{\mathcal{M}}(\Omega) \overset{!}{=} 0$, to determine bounds on different process equilibration times. This may motivate conditions on the generic Markovianization of process $\Upsilon$, or when a quantum process produces classical statistics, and what minimum times $\vec{T}$ would result in this. This addresses the question: how long does a quantum process take to lose memory? How coarse in time do measurements need to be to arrive at classical observed phenomena? And what extra assumptions on the system and dynamics are needed for this to occur [26]?

In the recent, related work [59], it is shown that for a large class of translation invariant Hamiltonians which satisfy a weak version of the ETH, two point correlation functions computed over a thermal state factorize for large times, with small deviations from this on average. Their setup therefore goes beyond the minimal one proposed in this work and the infinite time result of Ref. [25]. Interestingly, their results imply a kind of weak Markovianization, in that the factorization of thermal, temporal correlations implies that no memory is carried between

observables at two different times. This shows the kind of extra physical assumptions that may lead to emergent, general Markovianization and thermalization. Indeed, we expect that results presented here may lead to a generalization of those presented in Ref. [59] to arbitrary multitime correlations and to finite time intervals, which in turn could stimulate new insight regarding the question of Markovianization. It should also be noted that there are known cases where quantum non-Markovianity is only seen for three or higher order correlations [64], and that non-Markovianity can be hidden for arbitrarily long times [65, 66].

One could similarly investigate the times needed for a process to look classical. Both of these cases are physically relevant, as macroscopically it is highly typical to observe Markovian [67, 68] and classical phenomena. This would be an interesting avenue for further investigation, and the methods used here may be used to address parallel questions to the contemporary issue of equilibration time scales [46, 50–55].

## Acknowledgments

ND is supported by an Australian Government Research Training Program Scholarship and the Monash Graduate Excellence Scholarship. PS acknowledges financial support from a fellowship from "la Caixa" Foundation (ID 100010434, fellowship code LCF/BQ/PR21/11840014) and from the Spanish Agencia Estatal de Investigación (project no. PID2019-107609GB-I00), the Spanish MINECO (FIS2016-80681-P, AEI/FEDER, UE), and the Generalitat de Catalunya (CIRIT 2017-SGR-1127). KM is supported through Australian Research Council Future Fellowship FT160100073, Discovery Project DP210100597, and the International Quantum U Tech Accelerator award by the US Air Force Research Laboratory.

## A Proof of Eq. (38)

This proof is essentially reproduced from the Appendix of Ref. [25].

To prove Eq. (38), we require to further specify the process $\Upsilon$ to encompass a pure initial state, and only allow pure instruments (a pure instrument is one which preserves the purity of the input state). This allows us to argue that the evolution is according to an effective degenerate Hamiltonian. We will show that this generalizes to mixed states and instruments in the following.

Intuitively, more mixing means more statistical uncertainty, and so mixed instruments/states may only further equilibrate the system. Precisely, $\overline{|\langle \mathbf{A} \rangle_{\Upsilon} - \langle \mathbf{A} \rangle_{\Omega}|^2}^{\vec{T}}$ is convex in mixtures of pure instruments/states, so any bound for pure instruments/states may be used in a straightforward manner to produce a bound for mixed instruments/states. Therefore, we consider only exclusively pure instruments/states. This allows us to specify that the evolution is equivalently according to a non-degenerate (rank 1) Hamiltonian, $H' = \sum E_n |n\rangle \langle n|$, where $\{|n\rangle\}$ could be different for different 'steps' in the process. This is done by specifying a basis for each unitary evolution such that only one basis state $|n\rangle$ of any degenerate subspace may overlap with the $SE$ state, for each distinct energy.

First, consider Eq. (38) for a single sum over $n_i \neq m_i$. Expanding the instrument in its Kraus representation $\mathcal{A}(\cdot) \equiv \sum_\beta K^\beta(\cdot)K^{\beta\dagger}$, for any density operator $\sigma$,

$$
\begin{aligned}
\sum_{n_i \neq m_i} \left| \text{tr}[\mathcal{A}\mathcal{P}_{n_i m_i}(\sigma)] \right|^2 &= \sum_{n_i \neq m_i} \left| \sum_\beta \text{tr}[K^\beta(|n_i\rangle\langle n_i| \sigma |m_i\rangle\langle m_i|)K^{\beta\dagger}] \right|^2 \\
&= \sum_{n_i \neq m_i} |\text{tr}[A |n_i\rangle\langle n_i| \sigma |m_i\rangle\langle m_i|]|^2 \\
&= \sum_{n_i \neq m_i} |\sigma_{n_i m_i}|^2 \langle m_i| A |n_i\rangle\langle n_i| A^\dagger |m_i\rangle \\
&\leq \sum_{n_i \neq m_i} \sigma_{n_i n_i} \sigma_{m_i m_i} \langle m_i| A |n_i\rangle\langle n_i| A |m_i\rangle \\
&\leq \sum_{n_i, m_i} \text{tr}\left[ A\sigma_{n_i n_i} |n_i\rangle\langle n_i| A\sigma_{m_i m_i} |m_i\rangle\langle m_i| \right] \\
&= \text{tr}[A\$_i(\sigma)A\$_i(\sigma)] \\
&\leq \sqrt{\text{tr}[A\$_i(\sigma)\$_i(\sigma)A]\,\text{tr}[\$_i(\sigma)AA\$_i(\sigma)]} \\
&= \sqrt{\text{tr}\left[AA(\$_i(\sigma))^2\right]\text{tr}\left[AA(\$_i(\sigma))^2\right]} \\
&\leq \|A^2\|_p \text{tr}\left[(\$_i(\sigma))^2\right] = \|A\|_p^2 d_{\text{eff}}^{-1}[\sigma],
\end{aligned}
\tag{A.1}
$$

where we have defined the POVM element $A := \sum_\beta K^{\beta\dagger}K^\beta = A^\dagger$, and the energy eigenstate decomposition $\sigma := \sum_{n_i, m_i} \sigma_{n_i m_i} |n_i\rangle\langle m_i|$. In the fifth line we have used the Cauchy-Schwarz inequality $|\sigma_{nm}|^2 \leq \sigma_{nn}\sigma_{mm}$, valid for any positive hermitian operator $\sigma$ (equality for pure states). In the sixth line we have added the (non-negative) terms where $m_i = n_i$ to the sum. In the penultimate line we have again used Cauchy-Schwarz, but for the Hilbert-Schmidt inner product, $\text{tr}[A^\dagger B] \leq \|A\|_{\text{HS}}\|B\|_{\text{HS}}$ with $\|A\|_{\text{HS}} := \sqrt{\text{tr}[A^\dagger A]}$ and noting that $A = A^\dagger$ and $\sigma = \sigma^\dagger$. Finally, we have used the identity $\text{tr}(XY) \leq \|X\|_p \text{tr}(Y)$ for positive operators $X$ and $Y$, and that operator norms satisfy $\|X^\dagger X\|_p = \|X\|_p^2$.

Next, proving Eq. (38) for two sums over $n_i \neq m_i$,

$$
\begin{aligned}
\sum_{\substack{n_1 \neq m_1 \\ n_2 \neq m_2}} \left| \text{tr}[\mathcal{A}_2 \mathcal{P}_{n_2 m_2} \mathcal{A}_1 \mathcal{P}_{n_1 m_1}(\rho)] \right|^2 &= \sum_{\substack{n_1 \neq m_1 \\ n_2 \neq m_2}} \left| \sum_\alpha \text{tr}\left[ \mathsf{A}_2 |n_2\rangle \langle n_2| K_1^\alpha |n_1\rangle \langle n_1| \rho |m_1\rangle \langle m_1| K_1^{\alpha\dagger} |m_2\rangle \langle m_2| \right] \right|^2 \\
&= \sum_{\substack{n_1 \neq m_1 \\ n_2 \neq m_2}} \sum_{\alpha,\beta} |\rho_{n_1 m_1}|^2 \langle m_2| \mathsf{A}_2 |n_2\rangle \langle n_2| K_1^\alpha |n_1\rangle \langle m_1| K_1^{\alpha\dagger} |m_2\rangle \langle m_2| K_1^\beta |m_1\rangle \langle n_1| K_1^{\beta\dagger} |n_2\rangle \langle n_2| \mathsf{A}_2 |m_2\rangle \\
&\leq \sum_{\substack{n_1, m_1 \\ n_2, m_2}} \sum_\alpha \rho_{n_1 n_1} \rho_{m_1 m_1} |\langle m_2| \mathsf{A}_2 |n_2\rangle|^2 \langle n_2| K_1^\alpha (|n_1\rangle \langle n_1|) K_1^{\alpha\dagger} |n_2\rangle \langle m_2| K_1^\alpha(|m_1\rangle \langle m_1|) K_1^{\alpha\dagger} |m_2\rangle \\
&\quad + \sum_{\substack{n_1, m_1 \\ n_2, m_2}} \sum_{\alpha \neq \beta} \rho_{n_1 n_1} \rho_{m_1 m_1} |\langle m_2| \mathsf{A}_2 |n_2\rangle|^2 \langle n_2| K_1^\alpha |n_1\rangle \langle m_1| K_1^{\alpha\dagger} |m_2\rangle \langle m_2| K_1^\beta |m_1\rangle \langle n_1| K_1^{\beta\dagger} |n_2\rangle \\
&\leq \sum_\alpha \sum_{n_2, m_2} |\langle m_2| \mathsf{A}_2 |n_2\rangle|^2 \langle n_2| K_1^\alpha(\omega_1) K_1^{\alpha\dagger} |n_2\rangle \langle m_2| K_1^\alpha(\omega_1) K_1^{\alpha\dagger} |m_2\rangle \\
&\quad + \max_{i,j} |(\mathsf{A}_2)_{ij}|^2 \sum_{\alpha \neq \beta} \sum_{\substack{n_1, m_1 \\ n_2, m_2}} \langle m_2| K_1^\beta(|m_1\rangle \langle m_1|) K_1^{\alpha\dagger} |m_2\rangle \langle n_2| K_1^\alpha(|n_1\rangle \langle n_1|) K_1^{\beta\dagger} |n_2\rangle \\
&= \sum_\alpha \sum_{n_2, m_2} (K_1^\alpha(\omega_1) K_1^{\alpha\dagger})_{n_2 n_2} (K_1^\alpha(\omega_1) K_1^{\alpha\dagger})_{m_2 m_2} \text{tr}\left[ \mathsf{A}_2 |n_2\rangle \langle n_2| \mathsf{A}_2 |m_2\rangle \langle m_2| \right] \\
&\quad + \max_{i,j} |(\mathsf{A}_2)_{ij}|^2 \sum_{\alpha \neq \beta} \text{tr}\left[ K_1^\beta K_1^{\alpha\dagger} \right] \text{tr}[K_1^\alpha K_1^{\beta\dagger}] \\
&= \sum_\alpha \text{tr}\left[ \mathsf{A}_2 \$_2(K_1^\alpha(\omega_1) K_1^{\alpha\dagger}) \mathsf{A}_2 \$_2(K_1^\alpha(\omega_1) K_1^{\alpha\dagger}) \right] \\
&\leq \sum_\alpha \|\mathsf{A}_2\|_p^2 \text{tr}\left[ \left( \$_2(K_1^\alpha(\omega_1) K_1^{\alpha\dagger}) \right)^2 \right] \\
&\leq \|\mathsf{A}_2\|_p^2 \text{tr}\left[ \left( \sum_\alpha \$_2(K_1^\alpha(\omega_1) K_1^{\alpha\dagger}) \right)^2 \right] = \|\mathsf{A}_2\|_p^2 d_{\text{eff}}^{-1}[\mathcal{A}_1(\omega_1)],
\end{aligned}
$$

$$(\text{A.2})$$

where in the third line we have again used that $|\rho_{n_1 m_1}|^2 \leq \rho_{n_1 n_1} \rho_{m_1 m_1}$, and also split the sums $\sum_{\alpha, \beta} = \sum_{\alpha \neq \beta} + \sum_\alpha \delta_{\alpha\beta}$, adding the (non-negative) extra terms $n_1 = m_1$ and $n_2 = m_2$s. In the antepenultimate line we have chosen an orthogonal (*canonical*) Kraus representation for $\mathcal{A}_1$, a minimal representation such that $\text{tr}[K_1^{\alpha\dagger} K_1^\beta] \propto \delta^{\alpha\beta}$ [45]. At that point we have a term of the form of the seventh line of Eq. (A.1), and so we use that result to arrive at the next inequality. In the final line we bring the sum inside by the linearity of the trace, and as $\sum |x_i|^2 \leq |\sum x_i|^2$ for positive $x_i$.

The combination of Eqs. (A.1) and (A.2) generalize directly to arrive at Eq. (38) for arbitrarily many sums over $n_i \neq m_i$.

# B  Proof of Eq. (40)

This proof is reproduced from Ref. [50].

Consider the matrix $\overline{\mathcal{G}}^{(\ell)}$ as defined in Eq. (33). Then

$$
\|\overline{\mathcal{G}}^{(\ell)}\| \leq \max_{n'_\ell m'_\ell} \sum_{n,m} |\overline{\mathcal{G}}^{(\ell)}_{n_\ell m_\ell n'_\ell m'_\ell}|, \tag{B.1}
$$

noting that

$$\overline{\mathcal{G}}^{(\ell)}_{n_\ell m_\ell n'_\ell m'_\ell} = \overline{\exp[-i\Delta t (E_{n_\ell} - E_{n'_\ell} - E_{m_\ell} + E_{m'_\ell})]}^{T_\ell}$$

$$= \begin{cases} 1, & \text{if } (E_{n_\ell} - E_{n'_\ell} - E_{m_\ell} + E_{m'_\ell}) = 0, \qquad \text{(B.2)} \\ \dfrac{\exp[i(E_{n_\ell} - E_{n'_\ell} - E_{m_\ell} + E_{m'_\ell})T] - 1}{i(E_{n_\ell} - E_{n'_\ell} - E_{m_\ell} + E_{m'_\ell})T}, & \text{otherwise.} \end{cases}$$

Now, the sum in Eq. (B.1) can be split into intervals of width $\epsilon$, such that there are at most $N(\epsilon)$ energy gaps $(n, m)$ satisfying,

$$(k + 1/2)\epsilon > E_{n_\ell} - E_{n'_\ell} - E_{m_\ell} + E_{m'_\ell} \geq (k - 1/2)\epsilon, \qquad \text{(B.3)}$$

for each $k \in \mathbb{Z}$. For $k = 0$, we just take $|\overline{\mathcal{G}}^{(\ell)}_{n_\ell m_\ell n'_\ell m'_\ell}| \leq 1$, giving the first term of the bound Eq. (40). If $k$ is non-zero, then $|E_{n_\ell} - E_{n'_\ell} - E_{m_\ell} + E_{m'_\ell}| \geq (|k| - 1/2)\epsilon$, and so considering that there are $d_\mathrm{H}(d_\mathrm{H} - 1)$ terms in the sum, we can use Eq. (B.2) to arrive at,

$$\sum_{n,m} |\overline{\mathcal{G}}^{(\ell)}_{n_\ell m_\ell n'_\ell m'_\ell}| \leq N(\epsilon) \left( 1 + 2 \sum_{k=1}^{d_\mathrm{H}(d_\mathrm{H} - 1)/2} \frac{2}{(k - 1/2)\epsilon T} \right). \qquad \text{(B.4)}$$

Now, due to convexity,

$$\sum_{n=2}^{d_\mathrm{H}(d_\mathrm{H} - 1)/2} \frac{1}{n - 1/2} \leq \int_1^{d_\mathrm{H}(d_\mathrm{H} - 1)/2} \frac{1}{x} dx = \ln\left( \frac{d_\mathrm{H}(d_\mathrm{H} - 1)}{2} \right), \qquad \text{(B.5)}$$

and so

$$\sum_{n=1}^{d_\mathrm{H}(d_\mathrm{H} - 1)/2} \frac{1}{n - 1/2} \leq 2 + \ln\left( \frac{d_\mathrm{H}(d_\mathrm{H} - 1)}{2} \right) \leq 2\log_2(d_\mathrm{H}). \qquad \text{(B.6)}$$

The final inequality can be checked explicitly for $d_\mathrm{H} = 2$ and $d_\mathrm{H} = 3$, and confirmed for higher $d_\mathrm{H}$ by comparing the derivatives of both sides. Using this in Eq. (B.4), we arrive at the bound Eq. (40).

## C  Eq. (49) extended to $k = 3$ time instruments

The proof for higher $k$ proceeds in the same way as way described in the main body for $k = 2$, with the three types of terms addressed individually in the Sections 3.2.1, 3.2.2, and 3.2.3 equivalently appearing in the expansion for $k > 2$. The only non-trivial aspect is the counting of the multiplicity of particular terms. However, the total multiplicity is $(2^k - 1)^2$, which is what we take to be relevant for our main results for arbitrary $k$ (found explicitly in the definition of

the constant $C_k$ in Eq. (24)). Expanding Eq. (31) for $k = 3$ we have,

$$
\langle \mathbf{A_k} \rangle_{\Upsilon - \Omega} \underset{(k=3)}{=} \overbrace{\sum_{\substack{n_1 \neq m_1 \\ n_2 \neq m_2 \\ n_3 \neq m_3}} \text{tr}\big[ \mathcal{A}_3 \mathcal{P}_{n_3 m_3} \mathcal{A}_2 \mathcal{P}_{n_2 m_2} \mathcal{A}_1 \mathcal{P}_{n_1 m_1}(\rho) \big]\big[ e^{-i\Delta t_3 (E_{n_3} - E_{m_3})} e^{-i\Delta t_2 (E_{n_2} - E_{m_2})} e^{-i\Delta t_1 (E_{n_1} - E_{m_1})} \big]}^{=:X}
$$

$$
+ \overbrace{\sum_{\substack{n_1 \neq m_1 \\ n_2 \neq m_2}} \text{tr}\big[ \mathcal{A}_3 \$_3 \mathcal{A}_2 \mathcal{P}_{n_2 m_2} \mathcal{A}_1 \mathcal{P}_{n_1 m_1}(\rho) \big]\big[ e^{-i\Delta t_2 (E_{n_2} - E_{m_2})} e^{-i\Delta t_1 (E_{n_1} - E_{m_1})} \big]}^{=:Y_1}
$$

$$
+ \overbrace{\sum_{\substack{n_1 \neq m_1 \\ n_3 \neq m_3}} \text{tr}\big[ \mathcal{A}_3 \mathcal{P}_{n_3 m_3} \mathcal{A}_2 \$_2 \mathcal{A}_1 \mathcal{P}_{n_1 m_1}(\rho) \big]\big[ e^{-i\Delta t_3 (E_{n_3} - E_{m_3})} e^{-i\Delta t_1 (E_{n_1} - E_{m_1})} \big]}^{=:Y_2}
$$

$$
+ \overbrace{\sum_{\substack{n_2 \neq m_2 \\ n_3 \neq m_3}} \text{tr}\big[ \mathcal{A}_3 \mathcal{P}_{n_3 m_3} \mathcal{A}_2 \mathcal{P}_{n_2 m_2} \mathcal{A}_1(\omega_1) \big]\big[ e^{-i\Delta t_3 (E_{n_3} - E_{m_3})} e^{-i\Delta t_2 (E_{n_2} - E_{m_2})} \big]}^{=:Y_3}
$$

$$
+ \underbrace{\sum_{n_1 \neq m_1} \text{tr}\big[ \mathcal{A}_3 \$_3 \mathcal{A}_2 \$_2 \mathcal{A}_1 \mathcal{P}_{n_1 m_1}(\rho) \big] e^{-i\Delta t_1 (E_{n_1} - E_{m_1})}}_{=:Z_1}
$$

$$
+ \underbrace{\sum_{n_2 \neq m_2} \text{tr}\big[ \mathcal{A}_3 \$_3 \mathcal{A}_2 \mathcal{P}_{n_2 m_2} \mathcal{A}_1(\omega_1) \big] e^{-i\Delta t_2 (E_{n_2} - E_{m_2})}}_{=:Z_2}
$$

$$
+ \underbrace{\sum_{n_3 \neq m_3} \text{tr}\big[ \mathcal{A}_2 \mathcal{P}_{n_3 m_3} \mathcal{A}_2(\omega_2) \big] e^{-i\Delta t_3 (E_{n_3} - E_{m_3})}}_{=:Z_3} .
$$

(C.1)

Then,

$$
\overline{|\langle \mathbf{A_k} \rangle_{\Upsilon - \Omega}|^2}^{\vec{T}} \underset{(k=3)}{=} \overline{X^2} + \overline{Y_1^2} + \overline{Y_2^2} + \overline{Y_3^2} + \overline{Z_1^2} + \overline{Z_2^2} + \overline{Z_3^2}
$$

$$
+ 2\text{Re}\Big\{ (\overline{Y_1 Z_1^*} + \overline{Y_2 Z_1^*}) + (\overline{Y_1 Z_2^*} + \overline{Y_3 Z_2^*})
$$

$$
+ (\overline{Y_2 Z_3^*} + \overline{Y_3 Z_3^*}) + \overline{X Y_1^*}
$$

$$
+ \overline{X Y_2^*} + \overline{X Y_3^*}
$$

$$
+ \overline{Z_1 Z_2^*} + \overline{Z_1 Z_3^*} + \overline{Z_2 Z_3^*}
$$

$$
+ (\overline{X Z_1^*} + \overline{Y_1 Y_2^*}) + (\overline{X Z_2^*} + \overline{Y_1 Y_3^*}) + (\overline{X Z_3^*} + \overline{Y_2 Y_3^*})
$$

$$
+ (\overline{Y_1 Z_3^*} + \overline{Y_2 Z_2^*} + \overline{Y_3 Z_1^*}) \Big\} .
$$

(C.2)

Here we have grouped terms that will have similar bounds, such that the order of these groups here will agree with the equations below. Note that in the last two lines, we have grouped terms that are equal (thus getting an additional multiplicity). The key thing to note is that after carefully examination, one can see that the bounds of the individual terms in the proof for $k = 2$ in Section 3.2 can be directly generalized to higher $k > 2$. We need only determine what type of terms appear in Eq. (C.2); "diagonal", "cross" or "off-diagonal". In particular, the terms in the first line of Eq. (C.2) are diagonal and so can be bound directly using the method

of Section 3.2.1; the second, third, fourth and sixth lines can all be bound using Section 3.2.2 as they are cross terms; and finally the fifth and seventh lines are off-diagonal and so can be bound using the methods of Section 3.2.3. Looking at a representative example of each type of term, the diagonal terms can be bounded as

$$
\begin{aligned}
\overline{X^2} &= \sum_{\substack{n_1^{(\prime)} \neq m_1^{(\prime)} \\ n_2^{(\prime)} \neq m_2^{(\prime)} \\ n_3^{(\prime)} \neq m_3^{(\prime)}}} \overline{\mathcal{G}}^{(1)}_{n_1 m_1 n_1' m_1'} \overline{\mathcal{G}}^{(2)}_{n_2 m_2 n_2' m_2'} \overline{\mathcal{G}}^{(3)}_{n_3 m_3 n_3' m_3'} f(\mathcal{P}_{n_3 m_3}, \mathcal{P}_{n_2 m_2}, \mathcal{P}_{n_1 m_1}) f(\mathcal{P}_{n_3' m_3'}, \mathcal{P}_{n_2' m_2'}, \mathcal{P}_{n_1' m_1'})^* \\
&\leq \|\overline{\mathcal{G}}^{(1)}\| \|\overline{\mathcal{G}}^{(2)}\| \|\overline{\mathcal{G}}^{(3)}\| \sum_{\substack{n_1 \neq m_1 \\ n_2 \neq m_2 \\ n_3 \neq m_3}} |f(\mathcal{P}_{n_2 m_2}, \mathcal{P}_{n_1 m_1})|^2 \\
&\leq g_1 g_2 g_3 \|A_3\|_p^2 d_{\text{eff}}^{-1}[\mathcal{A}_2(\omega_2)].
\end{aligned}
\tag{C.3}
$$

In the second line we have used the steps outlined below Eq. (36), and then finally used Eq. (38) together with (45) and (40). Note that we define the generalization of the shorthand function $f$ (Eq. (34)) as

$$
f(\mathcal{X}, \mathcal{Y}, \mathcal{Z}) := \text{tr}[\mathcal{A}_3 \mathcal{X} \mathcal{A}_2 \mathcal{Y} \mathcal{A}_1 \mathcal{Z}(\rho)],
\tag{C.4}
$$

with $\mathcal{X}, \mathcal{Y}, \mathcal{Z}$ being either a projector, dephasing, or the identity map. An example of how we can bound a cross term is

$$
\begin{aligned}
2\text{Re}\{\overline{XY_1^*}\} &= 2\text{Re}\Big\{ \sum_{\substack{n_1^{(\prime)} \neq m_1^{(\prime)} \\ n_2^{(\prime)} \neq m_2^{(\prime)} \\ n_3 \neq m_3}} \overline{\mathcal{G}}^{(1)}_{n_1 m_1 n_1' m_1'} \overline{\mathcal{G}}^{(2)}_{n_2 m_2 n_2' m_2'} \overline{G}^{(3)}_{n_3 m_3} f(\mathcal{P}_{n_3 m_3}, \mathcal{P}_{n_2 m_2}, \mathcal{P}_{n_1 m_1}) f(\$, \mathcal{P}_{n_2' m_2'}, \mathcal{P}_{n_1' m_1'})^* \Big\} \\
&\leq 2\overline{G}^{(2)}_{\max} \|\overline{\mathcal{G}}^{(1)}\| \sqrt{\sum_{\substack{n_1' \neq m_1' \\ n_2' \neq m_2'}} |f(\$, \mathcal{P}_{n_2' m_2'}, \mathcal{P}_{n_1' m_1'})|^2} \Bigg( \sqrt{\sum_{\substack{n_1 \neq m_1 \\ n_2 \neq m_2}} |f(\mathcal{I}, \mathcal{P}_{n_2 m_2}, \mathcal{P}_{n_1 m_1})|^2} \\
&\qquad\qquad + \sqrt{\sum_{\substack{n_1 \neq m_1 \\ n_2 \neq m_2}} |f(\$, \mathcal{P}_{n_2 m_2}, \mathcal{P}_{n_1 m_1})|^2} \Bigg) \\
&\leq g_1 g_2 (2s_1) \|A_{3:2}\|_p^2 d_{\text{eff}}^{-1}[\rho]_{\min}.
\end{aligned}
\tag{C.5}
$$

Here, in the second line we used the triangle inequality argument as described above Eq. (43), and in the final line we have applied the bound Eq. (38) together with the bounds of the prefactors: (45) and (40). Finally, an example of the method to bound an off-diagonal term is

$$
\begin{aligned}
2\text{Re}\{\overline{Y_1 Z_3^*}\} &\leq 2|\overline{f(\$, \$, \mathcal{U}_1 - \$)}^{T_1} \overline{f(\$, \mathcal{U}_2 - \$, \$)}^{T_2} \overline{f(\mathcal{U}_3 - \$, \$, \$)}^{T_3}| \\
&\leq \frac{\sqrt{g_1 \|A_{3:1}\|_p^2 \, g_2 \|A_{3:2}\|_p^2 \, g_3 \|A_3\|_p^2}}{d_{\text{eff}}[\rho]_{\min}},
\end{aligned}
\tag{C.6}
$$

using the single time equilibration result of Ref. [50], analogous to the explanation of Eq. (48).

Therefore, using these methods to bound all terms in Eq. (C.2), Eq. (49) generalizes to $(2^3 - 1)^2 = 49$ terms, with an extra multiplicity of 2 for each $s_\ell$ in the cross terms

$$
\begin{aligned}
\overline{|\langle \mathbf{A_k} \rangle_\Upsilon - \langle \mathbf{A_k} \rangle_\Omega|^2}^{\vec{T}} \underset{(k=3)}{\leq} & \left\{ \left(1 + g_1 + g_2 + g_1 g_2 \right) g_3 \|A_3\|_p^2 + \left(g_1 + 1\right) g_2 \|A_{3:2}\|_p^2 + g_1 \|A_{3:1}\|_p^2 \right. \\
& + 2\left( \left((2s_2) + (2s_3)\right) g_1 \|A_{3:1}\|_p^2 + \left((2s_1) + (2s_3)\right) g_2 \|A_{3:2}\|_p^2 \right. \\
& + \left((2s_1) + (2s_2)\right) g_3 \|A_3\|_p^2 + g_1 g_2 (2s_3) \|A_{3:2}\|_p^2 \\
& + g_1 g_3 (2s_2) \|A_3\|_p^2 + g_2 g_3 (2s_1) \|A_3\|_p^2 \\
& + \left( \sqrt{g_1 g_2 \|A_{3:1}\|_p^2 \|A_{3:2}\|_p^2} + \sqrt{g_1 g_3 \|A_{3:1}\|_p^2 \|A_3\|_p^2} + \sqrt{g_2 g_3 \|A_{3:2}\|_p^2 \|A_3\|_p^2} \right) \\
& + 2g_1 (2s_2)(2s_3) \|A_{3:1}\|_p^2 + 2g_2 (2s_1)(2s_3) \|A_{3:2}\|_p^2 + 2g_3 (2s_1)(2s_2) \|A_3\|_p^2 \\
& \left. \left. + 3\sqrt{g_1 g_2 g_3 \|A_{3:1}\|_p^2 \|A_{3:2}\|_p^2 \|A_3\|_p^2} \right) \right\} d_{\text{eff}}^{-1}[\rho]_{\min}.
\end{aligned}
\tag{C.7}
$$

To determine the total multiplicity, given that there are $(2^k - 1)^2$ total terms in the $k$-time generalization of Eq. (35), there is also an additional factor of 2 for every $s_\ell$ that appears due to the triangle inequality used in the proof. Taking the largest power out the front by introducing an additional inequality, we arrive at a total multiplicity bounded above by $4^k 2^{k-1} = 2^{3k-1}$. Recalling the definition (25) and that $\mathfrak{g} \geq 1$, we again introduce an additional inequality with $\mathfrak{g}^k$ as a common factor, and arrive at the definition of $C_k$.

## D  Proof of result 1

Consider that each $\mathbf{A_w} \in \mathcal{M}_k$ has outcomes $\{\vec{x}\} = \{(x_1, x_2, \ldots, x_w)\}$ corresponding to the instrument $\mathbf{A_{\vec{x}}}$, where $w \leq k$. We can then bound the time averaged operational diamond norm, as defined in Eq. (15),

$$
\begin{aligned}
\overline{D_\mathcal{M}(\Upsilon, \Omega)}^{\vec{T}} &= \frac{1}{2} \overline{\max_{\mathbf{A_w} \in \mathcal{M}_k} \sum_{\vec{x}} |\text{tr}[\mathbf{A_{\vec{x}}} (\Upsilon_\mathbf{w} - \Omega_\mathbf{w})]|}^{\vec{T}} \\
&\leq \frac{1}{2} \sum_{\mathbf{A_w} \in \mathcal{M}_k} \sum_{\vec{x}} \overline{|\langle \mathbf{A_{\vec{x}}} \rangle_\Upsilon - \langle \mathbf{A_{\vec{x}}} \rangle_\Omega|}^{\vec{T}} \\
&\leq \frac{1}{2} \sum_{\mathbf{A_w} \in \mathcal{M}_k} \sum_{\vec{x}} \sqrt{\overline{|\langle \mathbf{A_{\vec{x}}} \rangle_\Upsilon - \langle \mathbf{A_{\vec{x}}} \rangle_\Omega|^2}^{\vec{T}}} \\
&\leq \frac{1}{2} \sum_{\mathbf{A_w} \in \mathcal{M}_k} \sum_{\vec{x}} \sqrt{C_w(\epsilon T_{\min}) d_{\text{eff}}^{-1}[\rho]_{\min}} \\
&\leq \frac{1}{2} \sum_{\mathbf{A_w} \in \mathcal{M}_k} \sum_{\vec{x}} \sqrt{C_k(\epsilon, T_{\min}) d_{\text{eff}}^{-1}[\rho]_{\min}} \\
&= \frac{\mathcal{S}_\mathcal{M} \sqrt{C_k(\epsilon, T_{\min})}}{2\sqrt{d_{\text{eff}}[\rho]_{\min}}},
\end{aligned}
\tag{D.1}
$$

where in the fourth line we have used Eq. (24), and in the fifth that $w \leq k$. Note that this proof follows closely to one given in Ref. [22], and is effectively reproduced from Ref. [25].

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
