# Peer review of "Equilibration of Multitime Quantum Processes in Finite Time Intervals"

_SciPost Physics Core, doi:SciPost Phys. Core 6, 043 (2023)_

## Round 2 · Referee Report · Anonymous (Referee 1) · 2023-1-24

Strengths

  1. Rigorous analytical results on the timescale at which “multi-time processes” are effectively in equilibrium.

  2. Very general set-up.

  3. Well written paper, with mostly appropriate referencing.

Weaknesses

  1. The results follow from previous proofs in the literature, and there are no new techniques for analyzing quantum dynamics.

  2. Setting and results not completely well motivated from the conceptual point of view.

Report

In this work, the authors study the notion of multi-time processes, which can be understood as a series of quantum dynamics between which an undetermined measurement or intervention is placed. Here, they consider the case in which the dynamics are unitary and generated by a time independent Hamiltonian.

These are of interest because in the context of many body dynamics, it is expected that many interesting models reach an effective equilibrium state, in which they stay for very long times. The question typically analyzed in the literature is: let a state evolve for time t, at which we measure a given observable. Does the expectation value of that observable correspond to the equilibrium state? Moreover, when is that effective equilibrium reached?

This is a timely question and a fair amount of work along different lines has been devoted to it. This includes a series of analytical works including references [10-13,15,37-38,42-47]. The present paper changes the question slightly, by generalizing it to scenarios where multiple measurements at different time intervals can happen, as described by the formalism of process tensors. The paper is well written and structured, and the key results used are referenced accordingly. It perhaps misses some discussion or citations about the wider literature on quantum dynamics and thermalization beyond the aforementioned papers (which are on a specific line of work within it).

In a previous paper [14], the authors studied the same setting, and prove a bound on the fluctuations away from equilibrium that holds for all times, following the proof idea of [11-13]. Here, instead, they consider the fluctuations for finite times. They show that an extension of [42] holds in this case. The proof follows [42] straightforwardly, with the complication that now there are “k” times to consider, and the number of terms increases accordingly. The complication is dealt with through various groupings of terms and chains of inequalities. The main idea is to consider the cases of small “k” first, and then explain how they generalize.

The technical part of the paper is largely about finding the way to bound the quantities considered to the point that the results [42] can be applied. A perhaps interesting point is that this more general result allows them to consider a diamond norm distance and other nontrivial figures of merit, as opposed to just expectation values. My impression however is that this in itself is maybe not a major technical complication. As long as the quantity to be bounded depends on the oscillating frequencies one can likely apply the result of [42] without a lot of technical difficulty.

In that sense, the paper can perhaps be seen as a straightforward generalization of [42] (which, to the author’s credit, is appropriately acknowledged throughout the manuscript). It may be that considering these multi-time processes is a conceptually non-trivial point, but the authors do not really make it clear enough in which type of situations this could be particularly interesting.

In this setting, in which it is important to consider measurements at different times, why should we care about processes that have equilibrated? Is it because we want to know whether all those different measurements will yield the same outcome? Is it just because it allows us to get other figures of merit? That is, I do not find the motivation for merging of equilibration results and the process tensor formalism strong enough. Considering all this, I do not quite find that the paper satisfies the journal’s acceptance criteria.

I also do not understand why they decided to split [14] and this reference into two. The setting of both papers appears to be the same, the difference being that the present paper considers finite time bounds, as well as slightly more general figures of merit. I do not find the two sets of results different enough to grant two papers, and together they could have perhaps made for a slightly stronger message: that analytical equilibration bounds generalize to multi-time processes, as opposed to doing the “multi-time process analogue” of each analytical equilibration result separately. In any case this is likely better left to the author’s judgment, they might have reasons to do so.

Another important point is that [42] (or for that matter, any other general analytical attempt to that problem) is rather limited in practice. The authors here mention that the bound grows with the system dimension as log d, which can be easily compensated by the fact that the effective dimension scales like d (note that in a many body system d is exponential in system size). The problem, which the authors may have missed (or at least I have not seen discussed), is that the biggest issue with the bound in [42] is the quantity N(\epsilon), which counts the number of energy gaps in a given interval.

In a many body system with a typical spectrum, this number is expected to be exponential in system size, likely down to epsilons that are exponentially small in system size. To give a rough idea, in a many body system one has to “fit” exponentially many levels in a spectrum of polynomial width. Thus there are inevitably very narrow energy windows with exponentially many energy levels. The same discussion also holds for the energy gaps.

As such, it is very likely that the equilibration time predicted by [42] is exponential in system size. As the authors discuss, this is still likely much smaller than the recurrence time, which is something, but it is still a big overestimate of the equilibration timescales in most models. In “real systems” these are expected to be perhaps polynomial in system size (which can be understood, for instance, in terms of transport and the relaxation of hydrodynamic modes).

Other comments:

-Why can you not apply Chebyshev’s inequality to the result? I think you can, just not writing it in terms of distance to the average over [0,T] but to the long time average.

-If the equilibrated process is not “classical”, what is the significance of the distance to classicality bein equilibrated? This point towards the end, and its actual significance, does not seem very well explained, and from the discussion it is perhaps a bit difficult to understand the notion of classicality clearly.

Requested changes

If the authors think there is room for it, I think the paper could do with a better motivation for considering this setup, and a clearer explanation of the novelty with respect to previous works on equilibration (if there is). I cannot think of specific changes beyond that.

  • validity: high
  • significance: low
  • originality: low
  • clarity: good
  • formatting: good
  • grammar: excellent

Author:  Neil Dowling  on 2023-02-01  [id 3291]

(in reply to Report 1 on 2023-01-24)
Category:
remark
answer to question
reply to objection

Dear Referee,

We thank you for your in-depth review and suggestions. We will likely vastly improve the manuscript based off of this report, pending recommendation from the Editor-in-charge (particularly regarding points you made about motivation and discussion).

We would like to comment on some of your concerns:
1. “In this setting, in which it is important to consider measurements at different times, why should we care about processes that have equilibrated? Is it because…”
This work is not just a simple combination of the ideas of equilibration together with process tensor. We have tried to make this clear in sections IV and V. There is a fundamental significance in these results regarding multitime features of quantum dynamical systems, such as markovianization and the emergence of classicality. Measurement/interaction perturbs quantum systems, so it is highly non-trivial to ask, what is the mechanism of the emergence of these multitime properties? On the quantum level, a molecule will ‘remember’ if I apply laser pulse to it. However, chemically such a process is Markovian in practice. Why is that so? Our work helps address foundational questions beyond thermalization and single-time equilibration.

2. “I also do not understand why they decided to split [14] and this reference into two. The setting of both papers appears to be…”
This manuscript constitutes a follow-up work to Ref. [14]. It is more technical and general than [14], arguing that all geometric measures equilibrate in finite time intervals (which are still long compared to realistic models, as pointed out by the referee), with a focus in the discussion on the emergence of classicality from measurement statistics. We determined that we could communicate these results in the most effective way via keeping these results separate, with the more physical results (in the form of probability bounds) and the simpler concepts in Ref. [14], while the current paper generalizes to further interesting cases.

3. “As such, it is very likely that the equilibration time predicted by [42] is exponential in system size. As the authors discuss…”
This is true and warrants further comments in the paper. However, the strength of these results is that they are valid under extremely general conditions, from ‘realistic’ physical setups to highly atypical situations. This is in the spirit of the original equilibration results.

4. “Why can you not apply Chebyshev’s inequality to the result? I think you can, just not writing it in terms of distance to the average over [0,T] but to the long time average.”
In Chebyshev, both the expectation value and the variance need to be taken with respect to the same distribution. Therefore applying Chebyshev to the finite-times results would give a distance to a finite-times averaged process, rather than the equilibrium process \Omega.

5. “If the equilibrated process is not “classical”, what is the significance of the distance to classicality bein equilibrated?”
This is analogous to the question of the significance of equilibration when the equilibrated state is not thermal. Here, the strength of our result is that we show analytically that a whole range of multitime features, such as classicality in the multitime measurement statistics of a quantum system, are stationary on average. This is under extremely general conditions. As we argue in the discussion, much like thermalization, additional assumptions are necessary to show complete 'classicalization' – that this stationary quantity is (approximately) zero; that the quantum process is classical (see penultimate paragraph of section IV on page 11).

We hope this convincingly addresses some of the Referee’s concerns.

---

## Round 2 · Referee Report · Anonymous (Referee 2) · 2023-2-17

Report

The manuscript by Dowling et al. reports results on the equilibration of quantum processes in finite time intervals. The work addresses a rather natural extension of a paper by the same authors (Ref. [14]) in which equilibration in the infinite-time-interval limit was discussed. While the question addressed in the present paper is a rather straightforward generalisation, it requires different methods and strategies to make progress on the technical side. I consider the paper a worthwhile study that should be published in some form. I do have a number of comments and requests that I would like to see addressed. While most of these comments in the below list concern minor issues regarding the presentation or notation, the first item on the list, while possibly easy to resolve, is a bit more substantial.

Once these issues are resolved, the paper certainly merits publication in some journal. Whether a paper like this, which extends an earlier result in a natural way, meets the bar for publication in SciPost Physics, I find much harder to answer. SciPost's requirements of a "groundbreaking [...] discovery" or "breakthrough on a previously-identified and long-standing research stumbling block" is not quite what I see in the paper. But a solid piece of research that makes progress in an interesting direction.

Requested changes

(a) Section IV, Eq. (51): According to the text below the equation, the first inequality in that equation is obtained by using "that $\mathcal{E}_{\mathcal{M}}$ is a minimum over all processes in the restricted set $\mathcal{K}$". This implies that each of the two minimisation functionals in the second line of (51) can be upperbounded by $\mathcal{D}_{\mathcal{M}}$. But the second minimization comes with a minus sign, and I don't see why the DIFFERENCE of the two minimisations should be upperbounded by the difference given in the third line of (51). The fact that, by assumption, $\mathcal{E}_{\mathcal{M}}(\Upsilon) \geq \mathcal{E}_{\mathcal{M}}(\Omega)$ doesn't help either. What am I missing?

An additional comment: While in the second line of (51) a minimum is taken over $\Lambda_\Omega$, the third line now suddenly seems to depend on $\Lambda_\Omega$. If this is the case, I'd assume that what is meant here is that the third line holds for arbitrary $\Lambda_\Omega \in \mathcal{K}$. Is this correct? Then this must be stated explicitly!

(b) The title sounds as if the results were restricted to non-Markovian processes, which, to my understanding, is not the case. "Equilibration of Quantum Processes in Finite Time Intervals" would then be a more accurate description of the content of the paper.

(c) While most of the paper is well written and clear, I found the abstract quite confusing, and pretty much impossible to understand before having read the paper. The first sentence makes a statement that is wrong in its absoluteness. Only the second sentence resolves this. Start the abstract by saying "Under suitable conditions, quantum processes...". Then: "Sufficient conditions for this result to hold are multitime observables that are coarse grained...". The next sentences starts with "This dictates...", but I don't find it obvious what "this" here is referring to. The conditions mentioned in the previous sentence? Or the result mentioned in the first sentence? Same in the fourth sentence: "We show that this leads to...". Which "this"? There is quite some room to improve the clarity and readability of the abstract.

(d) Third paragraph of the Introduction: "This IS a foundational question..."

(e) Same paragraph: What does "significantly interacting energy eigenstates" mean? States aren't interacting, I'd say. And if one writes a Hamiltonian in its energy eigenbasis, then there are no interacting terms in that basis. Please explain!

(f) Eq. (4): The notation $\mathbb{1}_2$ is so commonly used for the identity operator on $\mathbb{C}^2$ that I was pretty confused by the equation. Please introduce the notation here.

(g) Eq. (5): Same here, please briefly explain $^{T_1}$ notation. Sure, one can figure it out from the context, but a reader-friendly comment would be appreciated.

(h) Below Eq. (13): Does "trace properties" here mean "trace preserving properties"? If so, the latter would be clearer.

(i) Right thereafter: "...the latter is crucial for ensuring the causality of the process." I am surprised that trace properties ensure causality. Is there an argument that could be given as to why this is the case?

(j) Eq. (14): The notation $_{\Upsilon-\Omega}$ needs some guesswork as to what is meant. Please define!

(k) Below Eq. (17): "...as the average indistinguishability...". Please specify what average is meant here. (Time average, I suppose!?).

(l) First sentence of Sec. III: dotted --> dashed

(m) Eq. (23): Define $^\vec{T}$ notation. I don't think $\vec{T}$ had been used before.

(n) Below Eq. (26): "Here, $\mathcal{S}_{\mathcal{M}}$ is the total combined number of outcomes for all instruments..." I find this unclear, especially the "total combined". Please give a mathematical expression for $\mathcal{S}_{\mathcal{M}}$, I'd hope this makes it clearer.

---

## Round 3 · Author Response

We thank the Referees for carefully reviewing the manuscript and providing useful suggestions and feedback. We believe that this has significantly improved the paper.

In particular, we have: fixed a range of insufficient explanations and typos, more elegantly explained the relevance and impact of the results, added further comparison to related work and the state-of-the-art, detailed how this work fits into and contributes in a unique way to the overarching research program, and added more critical discussion of the results.

We hope the Editor and Referees agree that the changes, as detailed below, make this work a suitable candidate for publication in SciPost Physics Core.

---

## Round 3 · List of Changes

- New abstract
- Updated paragraph 2 of introduction to include more background information on different approaches to the question of statistical mechanics from pure state quantum mechanics.
- Added more citations from across a range of research areas in quantum thermalization/equilibration, largely in the second paragraph on page 1 (Refs. [10-18]) and below Eq. (29) (Refs. [13,14,58,59,60]).
- Updated paragraph 3 of introduction to explain that multitime equilibration implies a robustness to perturbation.
- Added paragraph 4 to introduction explaining difference of our approach compared to conventional equilibration papers. In particular how classicalization and markovianization go beyond the usual approach to thermalization.
- Added an extended discussion below what is now Eq. (29), comparing our bound to other (mostly numerical) results on more specific (and physical) models, detailing when the bound is useful.
- The proof Eq. (53) has been expanded with extra details.
- The title of the paper has changed slightly: "Non-Markovian" -> "Multitime"
- Changed paragraph 2 of the Introduction: "This IS a foundational question..."
- Changed paragraph 2 of the Introduction: "significantly interacting energy eigenstates" to “many significant populations in the energy eigenbasis….”
- Changed labelling of Hilbert spaces from numbers to letters on pages 2 and 3.
- "T" transpose notation clarifed with a comment in the second paragraph on page 3.
- A comment has been added to page 4 explaining trace properties of process tensor, " that tracing over a final
output leg of a process means..."
- Added sentence clarifying notation below Eq. (14).
- Page 5: "average indistinguishability" -> "time-average indistinguishability" .
- First sentence of Sec. III: dotted --> dashed.
- Added Eq. (23) and surrounding details.
- Added Eq. (28) and surrounding details.
- Added relevant citation to recent work Ref. [26]
- Updated penultimate paragraph of the conclusion to better detail comparisons with related work.

---

## Editorial Decision

published